# Foundation Model Insights and a Multi-Model Approach for Superior Fine-Grained One-shot Subset Selection

**Zhijing Wan** [1 2]  **Zhixiang Wang** [3 4]  **Zheng Wang** [1 2]  **Xin Xu** [5]  **Shin'ichi Satoh** [4 3]

## Abstract

One-shot subset selection serves as an effective tool to reduce deep learning training costs by identifying an informative data subset based on the information extracted by an information extractor (IE). Traditional IEs, typically pre-trained on the target dataset, are inherently dataset-dependent. Foundation models (FMs) offer a promising alternative, potentially mitigating this limitation. This work investigates two key questions: (1) Can FM-based subset selection outperform traditional IE-based methods across diverse datasets? (2) Do all FMs perform equally well as IEs for subset selection? Extensive experiments uncovered surprising insights: FMs consistently outperform traditional IEs on fine-grained datasets, whereas their advantage diminishes on coarse-grained datasets with noisy labels. Motivated by these finding, we propose RAM-APL (RAnking Mean-Accuracy of Pseudo-class Labels), a method tailored for fine-grained image datasets. RAM-APL leverages multiple FMs to enhance subset selection by exploiting their complementary strengths. Our approach achieves state-of-the-art performance on fine-grained datasets, including Oxford-IIIT Pet, Food-101, and Caltech-UCSD Birds-200-2011.

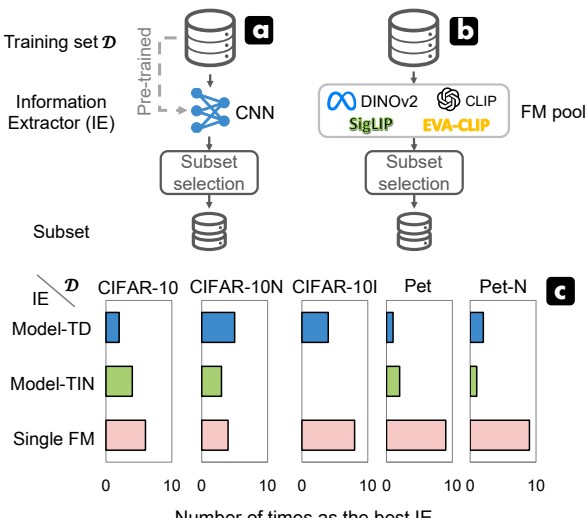

*Figure 1.* **Comparison of pipelines for one-shot subset selection.** (a) Traditional pipeline (He et al., 2024): Relies on a model pre-trained on the full training set of the target task to extract data information, but this leads to strong dataset dependency and additional pre-training cost. (b) Pipeline with a single foundation model (Xie et al., 2023): Replaces the small pre-trained model with a single FM, potentially mitigating dataset dependency. As shown in (c), on fine-grained datasets, using a single FM as an IE is significantly and consistently superior to using traditional IE, and improves the performance of subset selection at different sampling rates.

## 1. Introduction

Subset selection, also known as coreset selection (Zheng et al., 2023; Wan et al., 2024b), has become an effective approach to improve model training efficiency by identi-

[1]National Engineering Research Center for Multimedia Software, Institute of Artificial Intelligence, School of Computer Science, Wuhan University, China [2]Hubei Key Laboratory of Multimedia and Network Communication Engineering [3]The University of Tokyo, Japan [4]National Institute of Informatics, Japan [5]School of Computer Science and Technology, Wuhan University of Science and Technology, Wuhan 430065 China. Correspondence to: Zheng Wang <wangzwhu@whu.edu.cn>.

*Proceedings of the $42^{nd}$ International Conference on Machine Learning*, Vancouver, Canada. PMLR 267, 2025. Copyright 2025 by the author(s).

fying a small, representative subset of training data without significantly compromising model performance. This task is particularly important in scenarios involving large-scale datasets (Wan et al., 2024a; Wang et al., 2025; Jia et al., 2025), where full dataset training is computationally prohibitive. Subset selection methods can be broadly categorized into one-shot (Xia et al., 2024; Yang et al., 2024) and adaptive approaches (Karanam et al., 2022; Killamsetty et al., 2022). In this work, we focus on one-shot subset selection, which identifies subsets in a single pass, offering computational advantages over adaptive methods that require iterative selection during model training.

Traditional one-shot subset selection methods typically rely

on a pre-trained model as an information extractor (IE) to derive data characteristics such as features, gradients, or uncertainty scores. These characteristics are then used to identify the most representative subset. While numerous strategies—such as feature-based (Agarwal et al., 2020; Sener & Savarese, 2017), uncertainty-based (Coleman et al., 2019; Wu et al., 2024), and gradient matching-based approaches (Mirzasoleiman et al., 2020)—have been proposed, these methods fundamentally depend on pre-trained models obtained by training on the full dataset of the target task, as shown in Figure 1 (a). This inherently introduces significant dataset dependency, which limits their applicability, particularly in large-scale data scenarios. Efforts to reduce this dependency, such as employing lightweight proxy models (Coleman et al., 2019) or minimizing pre-training epochs (Guo et al., 2022), only partially mitigate the computational burden without fundamentally addressing the dataset dependency issue.

Recent advancements in foundation models (FMs), such as pre-trained vision models (Caron et al., 2021; Oquab et al., 2023) and vision-language models (Radford et al., 2021; Zhai et al., 2023; Sun et al., 2023), offer a promising alternative. A natural alternative to subset-based methods is fine-tuning or adapting FMs to the target dataset (Ding et al., 2023). While these approaches leverage pre-trained knowledge, they still require full-dataset access during fine-tuning, which undermines the computational efficiency that subset selection seeks to achieve. Moreover, these methods often face challenges such as overfitting on noisy datasets (Feng et al., 2024) and scalability issues on large datasets. In contrast, subset-based methods decouple the data selection process from task-specific training, enabling efficient learning without full-dataset reliance. With their robust generalization capabilities, FMs can serve as direct alternatives to traditional IEs, enabling dataset-agnostic subset selection pipelines, as illustrated in Figure 1 (b). Unlike traditional pipelines that rely on task-specific pre-training, FM-based pipelines eliminate the need for task-specific pre-training, making them well-suited for large and diverse datasets. Despite their potential, the advantages of FM-based pipelines over traditional methods remain under-explored. While some studies (Xie et al., 2023; Killamsetty et al., 2023) have investigated this approach, prior work (Xie et al., 2023) has revealed that simply using FMs for subset selection does not consistently lead to superior performance. This highlights critical open questions: Can FMs truly replace task-specific IEs in subset selection? If so, under what conditions?

In this paper, we conduct extensive experiments to investigate the strengths and limitations of using FMs as IEs for subset selection. Detailed experimental statistics and analysis can be found in *Single Model Study* section. Our experiments on subset selection using three kinds of models as IEs on five different types of image datasets, *i.e.,* CIFAR-

10 (Krizhevsky et al., 2009), CIFAR-10N-worse (CIFAR-10N) (Wei et al., 2022), CIFAR-10-imbalance (CIFAR-10I) (Cui et al., 2019), Oxford-IIIT Pet (Pet) (Parkhi et al., 2012)) and Oxford-IIIT Pet-N (Pet-N), revealed surprising findings: (1) FMs consistently outperform traditional IEs on both clean and noisy fine-grained datasets; and (2) FMs demonstrate limited advantages for subset selection on coarse-grained datasets with noisy labels.

While FMs are well-suited for fine-grained datasets, the optimal choice of FM as a feature extractor for subset selection remains an open question. Moreover, existing feature-based methods fail to comprehensively analyze feature distributions from both intra- and inter-class perspectives, resulting in suboptimal selection performance. To address these limitations, we introduce a novel subset selection pipeline that leverages multiple FMs with unknown selection performance to enhance fine-grained dataset selection. Our proposed RAM-APL method integrates diverse FMs (*i.e.,* DINOv2 and CLIP) and quantifies data importance through a systematic analysis of feature distributions across both intra- and inter-class levels, achieving state-of-the-art performance on three fine-grained image datasets.

The contributions of our work are three-fold:

- An in-depth study on the strengths and limitations of foundation models compared to traditional information extractors for subset selection reveals that foundation models consistently outperform traditional IEs on fine-grained datasets, whereas their advantage diminishes on coarse-grained datasets with noisy labels.

- A novel subset selection pipeline employing multiple foundation models with unknown selection performance as IEs is proposed for fine-grained image datasets. RAM-APL, an effective subset selection method, is designed based on the novel pipeline.

- Extensive experiments verify the superiority of RAM-APL on three fine-grained image datasets. Specifically on the Caltech-UCSD Birds-200-2011 dataset, RAM-APL achieves an average improvement of 6.4% in prediction accuracy over Random method across all sampling rates.

## 2. Related Works

Current one-shot subset selection methods typically follow a traditional selection pipeline, which consists of an information extractor, a measurer, and a selector. Various measures have been proposed to leverage the information provided by the extractor to assess data importance, including feature-based (Agarwal et al., 2020; Sener & Savarese, 2017), gradient-based (Kothawade et al., 2022; Killamsetty et al., 2021a), training dynamic-based (Toneva et al., 2018;

Swayamdipta et al., 2020; He et al., 2024; Zhang et al., 2024) and other weighting strategies (Zhou et al., 2020; Coleman et al., 2019; Zheng et al., 2022). Regardless of the above methods, their extractors are usually trained to converge on the full training set of the target task, rendering the pre-trained extractor data-dependent and limiting the applicability of subset selection to new large-scale datasets. For example, TDDS (Zhang et al., 2024) required 90 epochs of extractor training on ImageNet-1K to gather training dynamics, surpassing the 60 epochs needed for training the target model on the coreset. To solve this problem, Coleman *et al.* (Coleman et al., 2019) designed a small proxy model to perform data selection, achieving significantly faster pre-training. Guo *et al.* (Guo et al., 2022) proposed to pre-train a model for a small number of epochs. However, they do not break free from dataset dependency. Recently, some studies (Xie et al., 2023; Killamsetty et al., 2023) have explored using foundation models (FMs) as IEs for data selection, showing promise in addressing dataset dependency. Nevertheless, neither study has conclusively demonstrated that FMs outperform traditionally trained IEs. Specifically, (Xie et al., 2023) found that simply utilizing an FM does not guarantee superior data selection performance, raising questions about the viability of FMs as substitutes for traditional IEs. Our comprehensive investigation reveals that FMs universally dominate traditional IEs on fine-grained datasets (both clean and noisy), while their advantage diminishes on coarse-grained datasets with noisy labels. Furthermore, the contribution of an FM to subset selection varies across datasets. To maximize the potential of FMs for fine-grained subset selection, we propose strategically combining multiple FMs with complementary capabilities.

Since only features can be obtained from each FM, how to effectively use the unaligned features extracted from multiple FMs to measure and select data is the key problem. Existing feature-based subset selection methods can be classified into two main categories: geometry-based methods (Welling, 2009; Sener & Savarese, 2017; Xia et al., 2023) and decision boundary-based methods (Ducoffe & Precioso, 2018; Margatina et al., 2021). For geometry-based methods, studies (Welling, 2009; Sener & Savarese, 2017) selected samples whose distributions are not close to each other in feature space so that subsets do not have redundant information. These subsets usually make the model a good generalization. However, they treat samples whose distributions are not close to each other with equal importance, making subset selection for fine-grained datasets disregard inter-class distribution differences. Decision boundary-based methods select data close to the decision boundary, which is a time-consuming and biased selection process that is not beneficial for model generalization. Taking the best of both types of methods, we propose the subset selection method RAM-APL for fine-grained datasets.

## 3. Preliminary: Subset Selection

In downstream tasks such as image classification and recognition, we consider a large-scale training set $\mathcal{D} = \{I_1, \ldots, I_N\}$ with a dataset size N, where each sample $I_i = (x_i, y_i)$ consists of input data $x_i$ and its corresponding class label $y_i \in \{1, \ldots, C\}$. In scenarios where there's a specified budget $p$, subset selection is used to identify a subset $\mathcal{S}$ of $\mathcal{D}$ that contains the most informative data for the target downstream task. It is expected that the model $\theta^{\mathcal{S}}$ trained on $\mathcal{S}$ can perform on par with the model $\theta^{\mathcal{D}}$ trained on $\mathcal{D}$. The performance of subset selection is evaluated by the performance of model $\theta^{\mathcal{S}}$ on the test set of the target downstream task. The subset $\mathcal{S} = \{I_1, \ldots, I_M\}$ has a size $M$, where $M < N$, and the sampling rate for subset selection is defined as $p = M/N$. In the practical study, $p$ is pre-specified, and the subset $\mathcal{S}$ is selected with the expectation of maximizing the target model's accuracy while adhering to the budget constraint.

Subset selection relies on an Information Extractor (IE) to extract information from each sample, which is then used to assess the importance of the sample and select the most informative data. Traditionally, the IE is a model pretrained on the full training set, which inherently introduces dataset dependency, limiting the applicability of this approach across different datasets. To address this limitation, a more flexible and generalizable approach is necessary, and it is therefore crucial to explore alternatives that reduce or eliminate dataset dependency.

## 4. Single-Model Study

Foundation Models (FMs) have recently emerged as a promising alternative to traditional information extractors (IEs) for subset selection. However, the advantages of FM-based selection over conventional methods remain largely unexplored. In this section, we investigate whether a single foundation model can effectively replace traditional IEs and address the following two key questions: **Question 1:** In which cases are foundation models most effective, and in which cases are they not? **Question 2:** Do all FMs perform equally? Our extensive experiments reveal several key findings:

- **Observation 1:** FMs demonstrate limited advantages for subset selection on noisy, coarse-grained datasets.

- **Observation 2:** Conversely, FMs significantly and consistently outperform traditional IEs for subset selection on fine-grained datasets (both clean and noisy).

- **Observation 3:** Different FMs perform differently as information extractors for subset selection.

Inspired by Observations 2 and 3, we propose a FM-based

algorithm for superior fine-grained subset selection, which is elaborated in Section 5. In subsequent paragraphs, we provide detailed explanations for these observations.

**Experimental Setting**. To assess the applicability of foundation models as information extractors (IEs), we conducted subset selection experiments using a single model as the IE across five distinct image datasets: CIFAR-10 (Krizhevsky et al., 2009), CIFAR-10N-worse (CIFAR-10N) (Wei et al., 2022), CIFAR-10-imbalance (CIFAR-10I) (Cui et al., 2019), Oxford-IIIT Pet (Pet) (Parkhi et al., 2012)) and Oxford-IIIT Pet with 20% symmetric label noise (Oxford-IIIT Pet-N, abbreviated as Pet-N). We apply three kinds of models for feature extraction in subset selection respectively. Three kinds of models are: **(1)** models pre-trained on the target training dataset for ten epochs (Guo et al., 2022), referred to as model-TD. Once the target task changes, the model needs to be pre-trained again; **(2)** models pre-trained on Tiny-ImageNet (TIN) (Krizhevsky et al., 2012) for ten epochs, referred to as model-TIN. TinyImageNet is a larger classification dataset, models pre-trained on it possess a stronger representation ability compared to those pre-trained on target datasets. Given this, we think that model-TIN has the potential to serve as an alternative to traditional IEs without retraining when the target task changes; and **(3)** a single foundation model (*i.e.,* DINOv2, CLIP, SigLIP, or EVA-CLIP). To explore the impact of the above three kinds of models as IEs on selection algorithms, we implement four classical algorithms, *i.e.,* MIN, K-center Greedy (KCG) (Sener & Savarese, 2017), Graph Cut (GC) (Iyer et al., 2021) and Moderate_DS (MDS) (Xia et al., 2023) over the extracted features. Besides, we use each selection algorithm to select training samples with various sampling rates (*i.e.,* 10%, 30%, and 50%), and train target models over the selected subsets. We provide the detailed experimental setup and results in Appendix A.

We analyzed which of the three single models served as the most effective IE across four subset selection methods and three sampling rates. The frequency of each type of single model being the optimal IE under 12 settings on each dataset is presented in Figure 1 (c). Surprisingly, we found that directly using features extracted from the FM for subset selection does not consistently outperform features extracted from traditional pre-trained models.

(**Observation**) **FMs demonstrate limited advantages for subset selection on noisy, coarse-grained datasets. In contrast, FMs consistently outperform traditional IEs for subset selection on both clean and noisy fine-grained datasets.** In the case of selecting CIFAR-10N, the FM only emerged as the optimal IE in 4 out of 12 experimental setups. Conversely, the FM performed well on the other four datasets, especially on the Pet and Pet-N. For subset selection on CIFAR-10, the FM was the optimal IE in 6

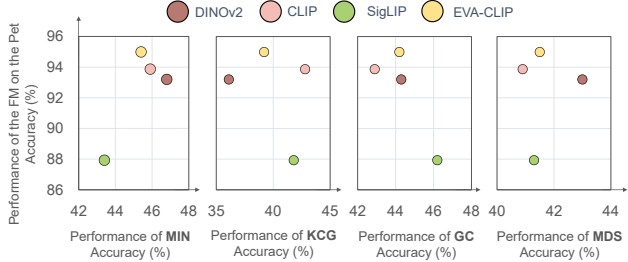

*Figure 2.* **Relationship between foundation model performance on the target task and subset selection performance using that FM as IE.** Superior target task accuracy does not necessarily lead to better subset selection performance across different foundation models and selection methods.

out of 12 experimental setups, but the best result at each sampling rate was achieved using model-TIN as the IE. In the case of CIFAR-10I, the FM was optimal in 8 out of 12 experimental setups, but at a low sampling rate of 1%, model-TD yielded the best results. Encouragingly, the single FM performed best in 9 out of 12 experimental setups on the Pet and Pet-N datasets and achieved the best results across all sampling rates. Thus, the FM presents a viable alternative to traditionally trained IEs for fine-grained image datasets. The Single-Model Study on more coarse- and fine-grained tasks shows the same conclusions, as summarized in Appendix A.2.

(**Observation**) **Different FMs perform differently as information extractors for subset selection, and the superior performance of FMs on downstream tasks does not guarantee better subset selection effects**. Various FMs are available, including DINOv2, CLIP, SigLIP, and EVA-CLIP. If the method is to be designed for fine-grained datasets according to the subset selection pipeline (b), an optimal FM needs to be identified first as the IE. An intuitive idea is to identify the optimal FM by testing each on the target task, with the best-performing FM chosen as the IE. However, we observe that superior performance on the downstream task does not guarantee better subset selection. As shown in Figure 2, although EVA-CLIP has strong zero-shot classification on Pet, it is not optimal for any selection method. Furthermore, our experiments indicate that the optimal FM as the IE varies depending on factors such as target datasets, selection methods, and sampling rates. For instance, Figure 2 demonstrates that for selecting 50% of the Pet dataset, DINOv2 performs best as the IE for the MIN method, while CLIP excels for the KCG method. Additional analysis of optimal FMs across sampling rates is presented in Appendix A.3. Therefore, Pipeline (b) requires an additional step to identify the best FM to achieve the most effective performance across different scenarios. This undoubtedly introduces an optimization detour, diverting focus from the primary goals of data measurement and selection.

While FMs are well-suited for fine-grained datasets, the optimal choice of FM as an feature extractor for FM-based subset selection remains an open question. Moreover, existing feature-based methods fail to comprehensively analyze feature distributions from both intra- and inter-class perspectives, resulting in suboptimal selection performance. To address these limitations, we explore a novel subset selection pipeline that directly employs multiple FMs with unknown individual contributions as IEs. Building on our pipeline, we propose the RAM-APL method, achieving state-of-the-art performance on multiple fine-grained datasets.

## 5. Proposed Method: RAM-APL

We are the first to investigate selection with multiple foundation models. In this section, we mainly propose a baseline method with multiple models as feature extractors. We introduce the problem formulation in Section 5.1. The subset selection method is then explained in detail in Section 5.2, which includes two metrics, namely ranking mean and accuracy of pseudo-class labels.

### 5.1. Problem Formulation

Multiple foundation models $\mathcal{M}_{\mathcal{F}}$ are used to extract information of training data in our method, where $\mathcal{M}_{\mathcal{F}} = \{M_F^1, \ldots, M_F^m\}$. Foundation models can be directly used as feature extractors, but features of the same samples extracted by different models are not aligned. Therefore, the two key challenges in our method design are effectively fusing features and accurately measuring sample importance based on the fused representations.

### 5.2. Method

The primary challenge in learning from fine-grained image datasets lies in their large intra-class differences and small inter-class differences. Existing subset selection methods either emphasize intra-class distribution while overlooking inter-class similarities or focus on decision-boundary samples while neglecting samples from other distributions within the class. To address these limitations, we propose RAM-APL, a selection method that quantifies data importance by jointly considering both intra-class and inter-class distributions.

**Feature Extraction** Given a fine-grained image dataset $\mathcal{D}$, we extract features using multiple FMs $M_F^i$, where $i \in \{1, \ldots, m\}$. The extracted feature set is denoted as $\mathcal{F} = [\mathcal{F}^1, \ldots, \mathcal{F}^m]$, where $\mathcal{F}^i = [F_1^i, \ldots, F_N^i]$ represents the feature representations of $\mathcal{D}$ obtained from the $i^{th}$ foundation model $M_F^i$. Each feature vector $F_j^i \in \mathbb{R}^{K_i}$ for a data sample $I_j$ is defined as: $F_j^i = \left[ f_j^{i,0}, f_j^{i,1}, \ldots, f_j^{i,K_i-1} \right] \in \mathbb{R}^{K_i}$, where $K_i$ represents the feature dimensionality of the $i^{th}$ model. Since FMs may produce features of varying dimensions, their representations are not necessarily aligned.

**RAnking Mean (RAM)** RAM maps features extracted by different foundation models from their unaligned feature spaces into a distance ranking space (an aligned space), facilitating the evaluation of data importance based on intra-class distribution.

After acquiring the feature set $\mathcal{F}$, we map the features extracted by each foundation model to a distance-ranking space. Specifically, given the feature set $\mathcal{F}^i = [F_1^i, \ldots, F_N^i]$ from foundation model $M_F^i$, we define the central feature of class $c$ as the mean feature vector:

$$\tilde{F}_c^i = \frac{1}{|S|} \sum_{j \in S} F_j^i, \tag{1}$$

where $S$ represents the set of indices belonging to class $c$. The Euclidean distance between a sample $F_j^i$ and its class center $\tilde{F}_c^i$ serves as a measure of representativeness, with smaller distances indicating higher representativeness (Xia et al., 2023):

$$d\left( F_j^i, \tilde{F}_c^i \right) = \| F_j^i - \tilde{F}_c^i \|_2, \tag{2}$$

where $\|\cdot\|_2$ denotes the Euclidean norm. Samples are ranked within each class according to their computed distances, producing ranked values $\mathcal{R}^i = [r_1^i, \ldots, r_{|S|}^i]$ for model $M_F^i$, where $r_j^i \in \mathbb{Z}^+$ and smaller values indicate closer distances. This process is repeated for all $m$ foundation models, mapping unaligned features into a unified distance-ranking space. The final ranking mean of class $c$ is denoted as:

$$\overline{\mathcal{R}}_c = [\overline{r}_1, \ldots, \overline{r}_{|S|}], \tag{3}$$

where $\overline{r}_j = \frac{1}{m*|S|} \sum_{i=1}^m r_j^i \in [0, 1]$ represents the normalized ranking mean for sample $I_j$. A smaller normalized ranking mean indicates greater alignment with class prototypes across foundation models. Visual analysis in Appendix B.4 further reveals that samples with lower normalized ranking means tend to exhibit more distinct target objects and simpler backgrounds.

**Accuracy of Pseudo-class Labels (APL)** APL maps features extracted by various foundation models from their unaligned feature space into a pseudo-class confidence score based on the unified inter-class distance ranking.

After obtaining the feature set $\mathcal{F}$, we assign pseudo-class labels to features extracted from each foundation model separately. Specifically, given the feature set $\mathcal{F}^i = [F_1^i, \ldots, F_N^i]$ from foundation model $M_F^i$, we first compute the central features of all $C$ classes using Equation (1), collectively denoted as $\tilde{F}^i = [\tilde{F}_0^i, \ldots, \tilde{F}_{(C-1)}^i]$. Next, we

calculate the Euclidean distances between each sample $F_j^i$ and all central features following Equation (2). These distances are represented as: $D(F_j^i) = [d_{j,0}^i, \ldots, d_{j,(C-1)}^i]$, where $d_{j,c}^i$ represents the distance between $F_j^i$ and the central feature $\tilde{F}_c^i$. The pseudo-class label for sample $I_j$ in the feature space of $M_F^i$ is then assigned based on the nearest central feature, computed as:

$$\tilde{y}_j^i = \arg\min D(F_j^i). \quad (4)$$

If the assigned pseudo-class label matches the ground-truth label, i.e., $\tilde{y}_j^i = y_j$, then the sample is considered correctly classified in the feature space of $M_F^i$, and we assign a score of $\varphi_j^i = 1$. Otherwise, we set $\varphi_j^i = 0$.

By repeating this process across all $m$ foundation models, we obtain a set of classification scores for each sample across different feature spaces. The average pseudo-class label accuracy for sample $I_j$ is then computed as:

$$\overline{\varphi}_j = \frac{1}{m} \sum_{i=1}^{m} \varphi_j^i. \quad (5)$$

A lower value of $\overline{\varphi}_j$ indicates that sample $I_j$ is more frequently misclassified across different feature spaces, suggesting a higher degree of similarity to other classes and thus greater difficulty in distinguishing it within the feature distribution. Finally, we represent the overall pseudo-class label accuracy for the entire dataset $\mathcal{D}$ as:

$$\overline{\varphi} = [\overline{\varphi}_1, \ldots, \overline{\varphi}_N]. \quad (6)$$

**Subset Selection** The importance of data samples in fine-grained learning is quantified through a linear combination of RAnking Mean and the Accuracy of Pseudo-class Labels (RAM-APL), formulated as:

$$Score = W_1 \times \overline{\mathcal{R}} + W_2 \times (1 - \overline{\varphi}). \quad (7)$$

Here, $W_1$ and $W_2$ control the contributions of intra-class and inter-class distributions, respectively. Inspired by (Swayamdipta et al., 2020), which highlights that easier samples facilitate optimization, we prioritize high intra-class similarity at lower sampling rates $p$, gradually incorporating harder samples as $p$ increases. Thus, $W_1$ and $W_2$ are dependent on the sampling rate $p$. The weight functions are defined as:

$$W_1 = \alpha + (1 - \alpha) \times \frac{1}{1 + e^{\beta*(p-0.5)}} \\ W_2 = 1 - W_1 \quad (8)$$

Samples with the smallest scores are selected into $\mathcal{S}$ up to the predefined budget. The hyper-parameters $\alpha$ and $\beta$ regulate the balance between intra-class and inter-class information across different sampling rates. Experimental results demonstrate that the best selection performance on fine-grained datasets is achieved using $\mathcal{M}_{\mathcal{F}} = \{CLIP, DINOv2\}$.

## 6. Experiments

### 6.1. Experimental Settings

**Datasets.** We evaluate our method on three classical fine-grained image classification datasets: Oxford-IIIT Pet (Pet) (Parkhi et al., 2012), Food-101 (Bossard et al., 2014), and Caltech-UCSD Birds-200-2011 (CUB-200-2011) (Wah et al., 2011). The Oxford-IIIT Pet comprises 7,349 images of 37 different breeds of cats and dogs. Food-101 has 101 classes, each with 750 training images and 250 test images. CUB-200-2011 consists of 11,788 images of 200 bird subcategories. Detailed dataset statistics are provided in Appedix A.1.

**Foundation Models as Feature Extractor.** We adopt two FMs as feature extractors for fine-grained image datasets, i.e., CLIP-VITl14 (Radford et al., 2021) and DINOv2-VITs14 (Oquab et al., 2023). The visual encoder of CLIP-VITl14 is used to extract image features, while the final layer [CLS] token embedding output of DINOv2-VITs14 serves as the feature representation. We emphasize that these FMs were not fine-tuned on the target datasets and were used solely as feature extractors for subset selection. The impact of varying the number of foundation models on selection performance is discussed in Section 6.4.

**Target Model Architecture & Training Parameters.** For Pet and Food-101 datasets, we use the 18-layer residual network (ResNet-18) (He et al., 2016) as the model backbone, initializing it randomly for training. For the CUB-200-2011 dataset, we adopt ResNet-50 as the model backbone, initialized with weights pre-trained on ImageNet (Deng et al., 2009). We follow the experimental setup from (Guo et al., 2022). Specifically, we use SGD as the optimizer with batch size 128, initial learning rate 0.1, Cosine decay scheduler, momentum 0.9, weight decay $5 \times 10^{-4}$, and 200 training epochs. For data augmentation, we employ a random resized crop to $224 \times 224$ resolution, followed by random horizontal flipping on training images. The code of our study is available at: https://github.com/ZhijingWan/RAM-APL.

**Evaluation Metric.** Prediction accuracy of a well-trained target model on the test set is used as the evaluation metric.

**Comparison Methods.** Multiple subset selection methods act as baselines for comparison. Specifically, we compare with (1) Random, which uniformly selects samples as the subset; (2) Herding (Welling, 2009); (3) K-Center Greedy (KCG) (Sener & Savarese, 2017); (4) Contextual Diversity (CD) (Agarwal et al., 2020); (5) Margin (Coleman et al., 2019); (6) Forgetting (Toneva et al., 2018); (7) GraNd (Paul et al., 2021); (8) Cal (Margatina et al., 2021); (9) Glister (Killamsetty et al., 2021b); (10) Graph Cut(GC) (Iyer et al., 2021); (11) Moderate_DS (MDS) (Xia et al., 2023); (12) MINimum distance (MIN), which selects samples with the minimum distance from the central feature of its class. Details of baselines are in Appendix B.1.

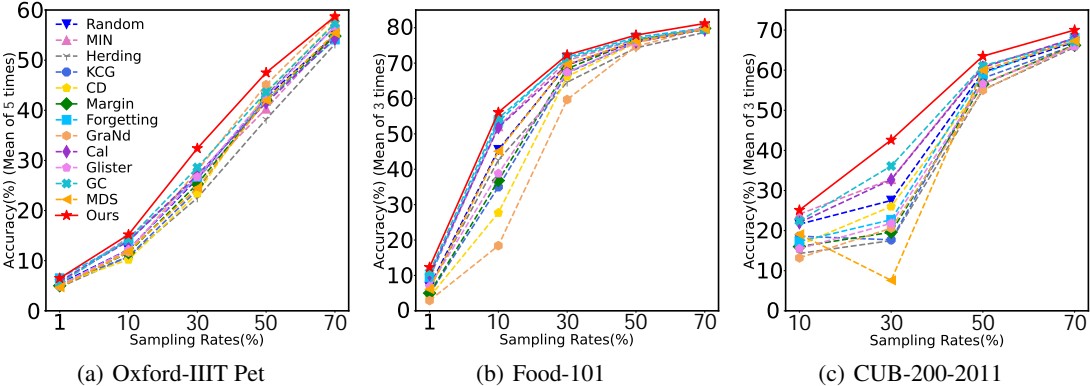

(a) Oxford-IIIT Pet  (b) Food-101  (c) CUB-200-2011

*Figure 3.* **Comparison of our method with baselines on three classical fine-grained image datasets.** Reported values correspond to mean accuracy.

We implemented each selection method based on the one-shot subset selection pipeline using code in the DeepCore library[1]. The information extractors used in baselines (2)-(12) were obtained using the traditional method, *i.e.,* training a model with the same backbone as the target model on the target training set for 10 epochs to ensure a fair comparison.

### 6.2. Comparison with Baselines

The results comparing the accuracy between the different subset selection methods on each fine-grained dataset are shown in Figure 3. Given each sampling rate, class-balanced sampling is performed. The experiments of each method on Pet were performed five times with different random seeds, while the experiments on Food-101 and CUB-200-2011 were performed three times with different random seeds due to the high computational effort. We adopt $\alpha = 0.2$ and $\beta = 1$ for our method across all datasets.

As shown in Figure 3, our method outperforms all baselines at each sampling rate. We compute the average performance gain of each method over Random across all sampling rates. On Pet, our method achieves a 3.74% average improvement, substantially outperforming the sub-optimal GC method, which shows a 1.52% average improvement. On Food-101, our gain reaches 4.44% compared to GC's 3.04%. On CUB-200-2011, our method shows a 6.40% average improvement versus GC's 2.78%. Detailed performance and additional cross-architecture generalization results are provided in Appendix B.

### 6.3. Ablation Study

Our method mainly consists of two novel designs: two feature measure metrics for multiple foundation models (*i.e.,* "RAM" and "APL"). We evaluate the effectiveness

---

[1]https://github.com/PatrickZH/DeepCore

*Table 1.* **Ablation study on the Pet dataset.** Model-TD refers to a model pre-trained on Pet for 10 epochs.

| Method | IE | Sampling rates | | |
|---|---|---|---|---|
| | | 1% | 50% | 70% |
| MIN | Model-TD | 5.6±0.7 | 40.3±2.6 | 55.2±2.7 |
| | CLIP | 5.6±0.2 | 45.9±1.8 | 56.3±0.7 |
| | DINOv2 | 6.2±0.1 | 46.8±2.0 | **60.5±2.9** |
| RAM | CLIP+DINOv2 | 5.9±0.3 | 47.1±1.4 | 56.5±2.7 |
| RAM-APL | | **6.5±0.4** | **47.5±1.9** | 58.7±2.2 |

of each design on the Pet dataset. Firstly, the RAM is designed primarily to effectively fuse the features extracted from multiple foundation models in terms of intra-class distribution, enabling the subset selection performance to be not inferior to that of any individual foundation model. As shown in Table 1, when using RAM to fuse the features extracted from CLIP and DINOv2 and selecting the samples with the minimum ranking mean, the performance of "RAM" is better than that of "MIN" with Model-TD or CLIP as the IE at each sampling rate. This validates the effectiveness of the RAM strategy. By combining APL and RAM to assess data importance for subset selection, our method outperforms the "MIN" baseline with DINOv2 as the IE at both 1% and 50% sampling rates. These results highlight the effectiveness of the joint RAM-APL strategy in fine-grained subset selection. Further analysis in Appendix B.5 shows that RAM-APL selects more diverse samples, enhancing overall coverage of the feature space.

### 6.4. Analysis and Discussion

**Parameter analysis.** The hyper-parameters $\alpha$ and $\beta$ are used to set the joint weights $W_1$ and $W_2$ according to Formula 8. We study the impact of them in Figure 4, testing five different values for $\alpha$ and $\beta$. In particular, we compared

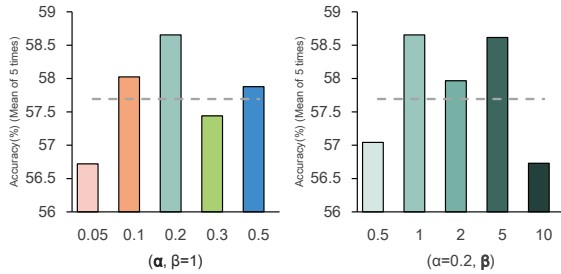

*Figure 4.* **Parameter analysis when sampling 70% of the Pet.** It shows that our method achieves the best performance when $\alpha = 0.2$ and $\beta = 1$. The grey dotted line indicates the selection method with score computed as $Score = \overline{\mathcal{R}} + (1 - \overline{\varphi})$, *i.e.,* a direct assignment of $W_1 = W_2 = 1$ without applying Formula 7.

the basic weight-setting strategy for fusion, *i.e.,* the equal-weighted fusion strategy, where $W_1 = 1$ and $W_2 = 1$. As illustrated in Figure 4, the best performance was achieved with $\alpha = 0.2$ and $\beta = 1$, outperforming the equal-weighted fusion strategy. When $\alpha = 0.2$ and $\beta = 1$, the fusion weights $(W_1, W_2)$ corresponding to 1%, 10%, 30%, 50%, and 70% sampling rates were (0.696, 0.304), (0.679, 0.321), (0.640, 0.360), (0.600, 0.400), (0.560, 0.44), respectively. As the sampling rate increases, $W_2$ increases while $W_1$ remains greater than $W_2$. This observation suggests that focusing more on inter-class feature distributions as the sampling rate increases helps to select better fine-grained subsets, but it is crucial to ensure that the intra-class assessment scores continue to dominate.

**Performance impact of the number of different FMs used for IE.** There exists a diverse set of FMs capable of extracting visual features, including DINOv2 (Oquab et al., 2023), CLIP (Radford et al., 2021), SigCLIP (Zhai et al., 2023) and EVA-CLIP (Sun et al., 2023). These models differ in their architectures, training strategies, and training datasets, leading to distinct knowledge and representation capabilities (as demonstrated in Appendix B.6). This raises a natural question: Does incorporating more FMs as IEs enhance our method's performance?

To explore this, we evaluate different combinations of DINOv2-VITs14, CLIP-VITL14, SigLIP-base-patch16-224, and EVA-CLIP-8B[2] on the Pet dataset. As shown in Table 2, using multiple FMs yields better overall performance than any single model. DINOv2+CLIP achieves the best trade-off between efficiency and accuracy, while adding EVA-CLIP yields further overall gains when computational resources permit. These findings support the benefit of multi-model consensus in our framework.

In the main experiments, we adopt DINOv2 and CLIP as our default IE pair, which yields consistent improvements over

---

[2]We use the SigLIP-base-patch16-224 and EVA-CLIP-8B from HuggingFace (Wolf et al., 2019)

*Table 2.* **Performance comparison of our method using different numbers of foundation models as information extractors.** Here, "D", "C", "S" and "E" represent DINOv2, CLIP, SigLIP, and EVA-CLIP, respectively.

| IE | | | | Sampling rates | | | | |
|---|---|---|---|---|---|---|---|---|
| D | C | S | E | 1% | 10% | 30% | 50% | 70% |
| ● | ○ | ○ | ○ | 5.9±0.3 | 15.4±1.1 | 31.6±2.3 | 47.7±1.1 | 57.9±4.1 |
| ○ | ● | ○ | ○ | 5.7±0.4 | 15.0±0.2 | 27.9±1.2 | 43.6±1.9 | 57.0±0.4 |
| ○ | ○ | ● | ○ | 6.6±0.3 | 14.1±1.0 | 28.8±1.1 | 43.9±1.7 | 55.1±2.6 |
| ○ | ○ | ○ | ● | 5.4±0.3 | 15.0±0.6 | 30.2±2.5 | 44.4±2.3 | 56.6±1.8 |
| ● | ● | ○ | ○ | 6.5±0.4 | 15.2±1.2 | 32.4±2.9 | 47.5±1.9 | **58.7±2.2** |
| ● | ○ | ● | ○ | 5.9±0.3 | 16.2±0.1 | 31.4±3.2 | 45.0±1.3 | 58.6±1.2 |
| ● | ○ | ○ | ● | 6.0±0.6 | 16.0±0.9 | **35.8±2.9** | 46.5±1.8 | 54.9±3.5 |
| ○ | ● | ● | ○ | 6.4±0.2 | 15.1±0.4 | 29.8±1.6 | 45.9±1.3 | 56.2±2.7 |
| ○ | ● | ○ | ● | 5.9±0.3 | 15.5±0.7 | 31.4±1.7 | 44.2±2.2 | 55.9±1.8 |
| ○ | ○ | ● | ● | **6.7±0.4** | 16.2±0.6 | 34.7±0.3 | 45.7±0.8 | 56.6±2.4 |
| ● | ● | ● | ○ | 6.2±0.8 | 15.6±0.5 | 33.2±1.4 | **48.3±1.1** | 57.6±0.1 |
| ● | ● | ○ | ● | 6.0±0.4 | **17.5±1.0** | 35.2±1.8 | 47.9±1.5 | 55.6±2.1 |
| ● | ○ | ● | ● | 6.1±0.3 | 16.8±0.6 | 34.4±2.1 | 47.0±2.0 | 55.1±1.6 |
| ○ | ● | ● | ● | 6.1±0.2 | 16.1±0.3 | 33.9±1.4 | 46.8±1.5 | 55.1±0.5 |
| ● | ● | ● | ● | 6.5±0.2 | 16.8±1.1 | 34.0±2.7 | 46.3±0.5 | 56.9±1.1 |

*Table 3.* **Comparison of feature fusion strategies.**

| Fusion strategy | Sampling rates | | | | |
|---|---|---|---|---|---|
| | 1% | 10% | 30% | 50% | 70% |
| Concatenate | 5.9±0.4 | **16.3±0.4** | 31.7±1.3 | **47.7±3.0** | 57.8±1.2 |
| Ours | **6.5±0.4** | 15.2±1.2 | **32.4±2.9** | 47.5±1.9 | **58.7±2.2** |

subset selection baselines across three fine-grained datasets.

**Feature fusion strategy.** Features extracted from different foundation models often exhibit misalignment due to architectural and training discrepancies. In RAM-APL, a simple fusion baseline is to concatenate features from different foundation models, referred to as "Concatenate." As shown in Table 3, our proposed fusion strategy outperforms simple concatenation, particularly under higher sampling ratios, which are critical in real-world deployment scenarios.

## 7. Conclusion

This work is the first to explore in-depth foundation models as information extractors (IEs) for one-shot subset selection. Our analysis revealed surprising insights: FMs consistently outperform traditional IEs on fine-grained datasets, whereas their advantage diminishes on coarse-grained datasets with noisy labels. Motivated by these findings, we developed the RAM-APL method, which integrates multiple FMs to enhance subset selection for fine-grained datasets. This pioneering integration paves the way for exploring vast frontiers of subset selection in the era of big data.

## Impact Statement

RAM-APL improves data efficiency in machine learning by leveraging multiple foundation models for one-shot subset selection, particularly benefiting fine-grained classification scenarios. By reducing reliance on large, fully labeled datasets, RAM-APL has the potential to lower the barriers to deploying high-performing models in low-resource or underrepresented domains. However, as with all data selection techniques, careful consideration is needed to avoid reinforcing dataset biases or unintentionally omitting critical minority samples. Our method does not involve synthetic data generation or human subjects and poses no direct ethical risks.

## Acknowledgment

We thank the anonymous reviewers for their valuable feedback. This work is partly supported by JSPS KAKENHI Grant Number JP23K24876, JST ASPIRE Program Grant Number JPMJAP2303 and the National Natural Science Foundation of China under Grant 62376201.

Work was done during Zhijing Wan's internship at the National Institute of Informatics, Japan.

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

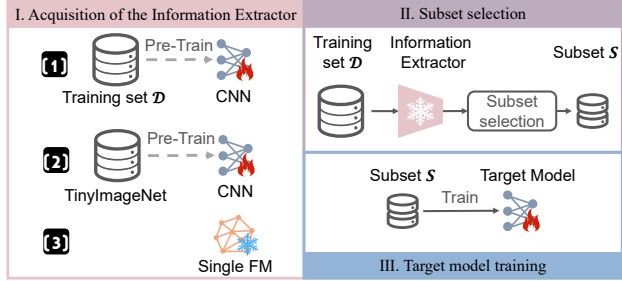

*Figure 5.* Framework of the Single-Model Study.

## A. More Details on the Single-Model Study

To investigate whether foundation models (FMs) can serve as alternatives to traditional information extractors (IEs), we conducted a comprehensive subset selection study using a single model as the IE across nine image classification datasets, uncovering surprising insights. The findings were derived primarily based on results from five different types of datasets (*i.e.,* CIFAR-10 (Krizhevsky et al., 2009), CIFAR-10N-worse (Wei et al., 2022), CIFAR-10-imbalance (Cui et al., 2019), Oxford-IIIT Pet (Parkhi et al., 2012) and Oxford-IIIT Pet with 20% symmetric label noise), with results from four additional datasets (*i.e.,* CIFAR-100 (Krizhevsky et al., 2009), Caltech-UCSD Birds-200-2011 (CUB-200-2011) (Wah et al., 2011), CIFAR-10 with 20% symmetric label noise, and Oxford-IIIT Pet with 40% symmetric label noise) serving as a strengthened proof of the findings to ensure rigor. To adequately compare with traditional IEs, we evaluated various FMs, including a vision foundation model (*i.e.,* DINOv2 (Oquab et al., 2023)) and vision-and-language foundation models (*i.e.,* CLIP (Radford et al., 2021), SigLIP (Zhai et al., 2023) and EVA-CLIP (Sun et al., 2023)). Besides, we implement four classical subset selection methods, *i.e.,* MIN, K-center Greedy (KCG) (Sener & Savarese, 2017), Graph Cut (GC) (Iyer et al., 2021), and Moderate_DS (MDS) (Xia et al., 2023) over the extracted features. Our study follows the framework shown in Figure 5. Details and results of the study are provided below.

### A.1. Experimental Settings

**Dataset Details**    As summarized in Table 4, We adopted nine datasets with different characteristics for the Single-Model Study:

**(1) CIFAR-10**: Contains 60,000 images of $32 \times 32 \times 3$ size in 10 classes, with 6,000 images per class. The dataset is split into 50,000 training images and 10,000 test images;

**(2) CIFAR-10 with 20% symmetric label noise (CIFAR-10N-0.2Sym)**: Adds 20% symmetric label noise to the original CIFAR-10 dataset;

*Table 4.* Dataset statistics. "Task" indicates whether the dataset was used in the Single-Model Study (Single) or the Multi-Model-based Subset Selection task (Multi).

| Dataset | Train | Test | Classes | Noise | Task |
|---|---|---|---|---|---|
| CIFAR-10 | 50,000 | 10,000 | 10 | - | Single |
| CIFAR-10N-0.2Sym | 50,000 | 10,000 | 10 | 20% | Single |
| CIFAR-10N | 50,000 | 10,000 | 10 | 40.21% | Single |
| CIFAR-10I | 20,431 | 10,000 | 10 | - | Single |
| CIFAR-100 | 50,000 | 10,000 | 100 | - | Single |
| Pet | 3,680 | 3,669 | 37 | - | Single&Multi |
| Pet-N | 3,680 | 3,669 | 37 | 20% | Single |
| Pet-N-0.4Sym | 3,680 | 3,669 | 37 | 40% | Single |
| CUB-200-2011 | 5,994 | 5,794 | 200 | - | Single&Multi |
| Food-101 | 75,750 | 25,250 | 101 | - | Multi |

*Table 5.* Characteristics of image datasets.

| Datasets | Grained Level | Noise label | Imbalance |
|---|---|---|---|
| CIFAR-10 | Coarse | ✗ | ✗ |
| CIFAR-10N | Coarse | ✓ | ✗ |
| CIFAR-10I | Coarse | ✗ | ✓ |
| CIFAR-100 | Coarse | ✗ | ✗ |
| Oxford-IIIT Pet | Fine | ✗ | ✓ |
| Oxford-IIIT Pet-N | Fine | ✓ | ✓ |
| CUB-200-2011 | Fine | ✗ | ✗ |

tractor was evaluated by the prediction accuracy of models trained on the selected subsets.

**Subset Selection Methods.** We employ multiple subset selection methods utilizing feature information for the Single-Model Study:

**(1) MINimum distance (MIN):** Selects samples closest to the central feature of their class;

**(2) K-center Greedy (KCG)** (Sener & Savarese, 2017): Selects a budget-sized subset $\mathcal{S}$ to minimize the largest distance between any sample in $\mathcal{D}\backslash\mathcal{S}$ and its closest sample in $\mathcal{S}$;

**(3) Moderate_DS (MDS)** (Xia et al., 2023): Selects samples with scores near the median of all samples, where scores represent distances to their class central feature;

**(4) Graph Cut (GC)** (Iyer et al., 2021): Greedily selects samples that maximize the Graph Cut function constructed using their features.

**Target Model Architecture & Training Parameters.** For datasets (1)-(8), we use an 18-layer residual network (ResNet-18) (He et al., 2016) as the model backbone, initializing it randomly for pre-training. For the CUB-200-2011 dataset, we adopt ResNet-50, initialized with weights pre-trained on ImageNet. We use SGD as the optimizer with batch size 128, initial learning rate 0.1, Cosine decay scheduler, momentum 0.9, weight decay $5 \times 10^{-4}$, and 200 training epochs. For data augmentation, we apply the random crop with 4-pixel padding and random flipping on the $32 \times 32$ training images of datasets (1)-(5). For datasets (6)-(9), we employ a random resized crop to a resolution of $224 \times 224$, followed by random horizontal flipping. Prediction accuracy is used as the evaluation metric, and the results of each method are averaged over three independent selections with different random seeds.

**(3) CIFAR-10N-worse (CIFAR-10N):** Maintains the same structure as CIFAR-10 but includes 40.21% human-annotated real-world noisy labels;

**(4) CIFAR-10-imbalance (CIFAR-10I):** Constructed using the method in (Cui et al., 2019) to create a long-tailed distribution. The number of samples in class $k$ was set to $n_k \times \mu^k$, where $n_k$ is the original count. The imbalance ratio $\rho$ is defined as $\rho = \frac{\max_k\{n_k\}}{\min_k\{n_k\}}$. In our experiments, we set $\rho = 10$. The training sample counts for classes 0 through 9 are 5,000, 3,871, 2,997, 2,320, 1,796, 1,391, 1,077, 834, 645, and 500, respectively. The original test set is used for evaluation;

**(5) CIFAR-100:** Contains 60,000 images of $32 \times 32 \times 3$ size in 100 classes, with 500 training images and 100 test images per class;

**(6) Oxford-IIIT Pet (Pet):** Includes 7,349 images in 37 classes, with 3,680 training images and 3,669 test images. The dataset is imbalanced;

**(7) Oxford-IIIT Pet with 20% symmetric label noise (Pet-N):** Adds 20% symmetric label noise to the original Pet dataset;

**(8) Oxford-IIIT Pet with 40% symmetric label noise (Pet-N-0.4Sym):** Adds 40% symmetric label noise to the original Pet dataset;

**(9) Caltech-UCSD Birds-200-2011 (CUB-200-2011):** Contains 11,788 images in 200 classes, with approximately 30 images per class for training.

Under symmetric noise, true labels are randomly reassigned to other classes. Table 5 shows characteristics of each kind of dataset.

**Single Foundation Model.** Our experiments employed DINOv2-ViTs14, CLIP-ViTl14, SigLIP-base-patch16-22, and EVA-CLIP-8B as the information extractor, respectively. Notably, these FMs were not fine-tuned on the target datasets and were solely used for feature extraction during subset selection. The selection performance of each ex-

## A.2. When Are FMs Most Effective, and When Are They Not?

To answer this question, we conducted a comprehensive Single-Model Study and derived two key insights, primarily

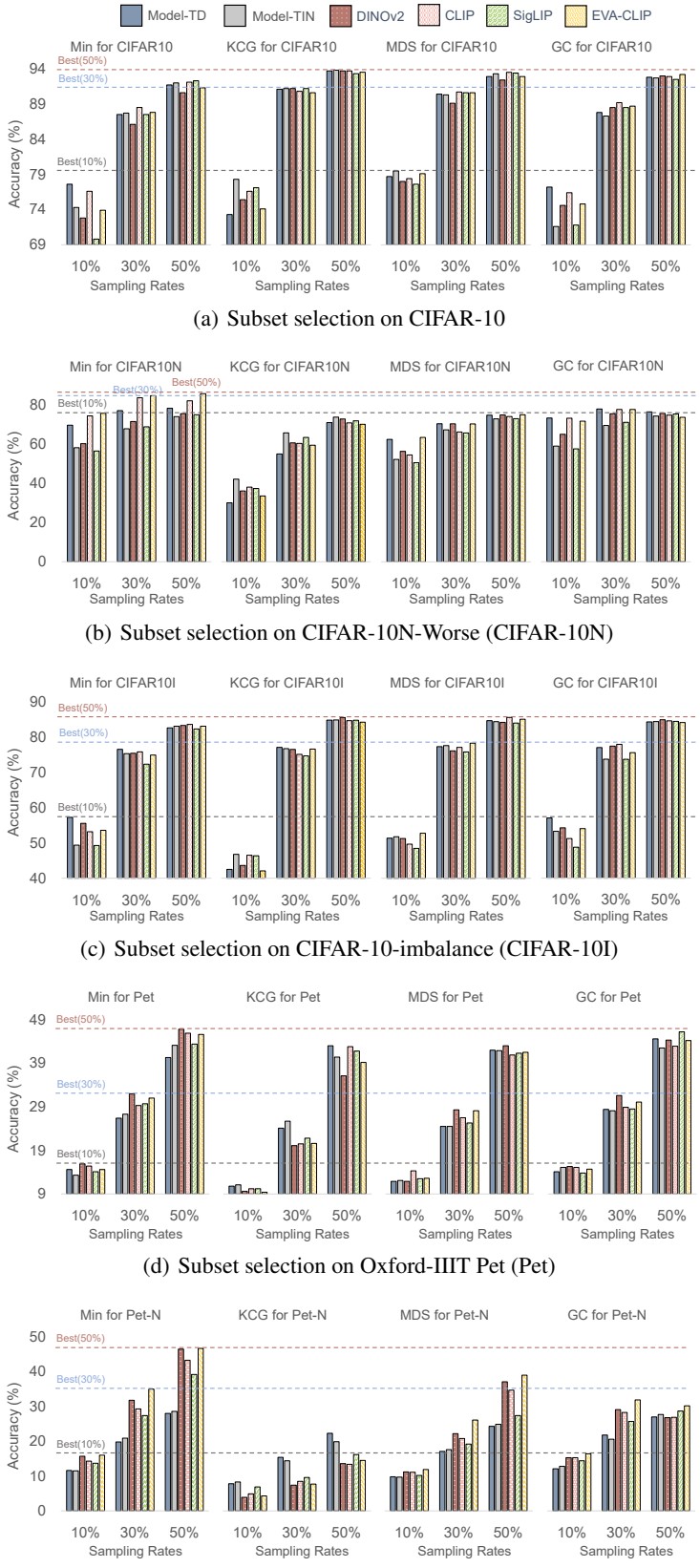

(a) Subset selection on CIFAR-10

(b) Subset selection on CIFAR-10N-Worse (CIFAR-10N)

(c) Subset selection on CIFAR-10-imbalance (CIFAR-10I)

(d) Subset selection on Oxford-IIIT Pet (Pet)

(e) Subset selection on Oxford-IIIT Pet with 20% symmetric label noise

*Figure 6.* Single-model study on five Target Datasets (TDs). Best viewed in color.

based on Figure 6. After analyzing which single model performs best as the IE when sampling datasets with different rates and subset selection methods, we surprisingly found that directly using features extracted from the FM for subset selection does not always outperform features extracted from traditional pre-trained models. Specifically, (1) FMs consistently outperform traditional IEs on both clean and noisy fine-grained datasets; and (2) FMs demonstrate limited advantages for subset selection on coarse-grained datasets with noisy labels. Additionally, while Model-TIN shows potential as an alternative to traditional IEs on CIFAR-10, its utility does not generalize to other scenarios.

These insights are further validated by additional results on clean datasets (Figure 7) and datasets with noisy labels (Figure 8), reinforcing the rigor of our findings. Specifically, as shown in Figure 7, the FM serves as the optimal IE in 7 out of 12 setups for CIFAR-100, but the best result at 10% and 50% sampling rates are achieved using model-TD as the IE. In comparison, the FM is optimal in 9 out of 12 setups for CUB-200-2011, consistently achieving the best results across all sampling rates. As shown in Figure 8, while FMs are the optimal IE in 9 out of 12 setups on CIFAR-10N-0.2Sym, model-TD achieves the best result at the 10% sampling rate. In comparison, on Pet-N-0.4Sym, despite 40% symmetric label noise, FMs are the optimal IE in 9 out of 12 setups. More importantly, FMs achieve the best performance across all sampling rates, significantly outperforming results obtained with model-TD.

### A.3. Do All FMs Perform Equally?

The unexpected findings motivated our design of a more effective subset selection method for fine-grained image datasets, leveraging the strengths of FMs as IEs. Given the availability of multiple FMs, designing a single-FM-based subset selection pipeline requires identifying the optimal FM for the task. However, as illustrated in Figure 6(d), the performance of FMs as IEs varies significantly depending on the sampling rate and subset selection algorithm. For example, under the KCG method, SigLIP achieves the best performance at the 30% sampling rate, while CLIP is optimal when sampling 50% of Pet. In contrast, when using the MIN method at 50% of Pet, DINOv2 emerges as the best IE, whereas GC favors SigLIP. These observations highlight that FMs do not perform equally across different scenarios.

## B. More Details on Multi-Model Selection

### B.1. Comparison Methods

We compare with 12 subset selection methods:

**(1) Random**: Uniformly selects samples to form the subset;

**(2) Herding** (Welling, 2009): Greedily selects samples to minimize the distance between the subset center and the full training dataset center in the feature space;

**(3) K-Center Greedy (KCG)** (Sener & Savarese, 2017): Selects a budget-sized subset $\mathcal{S}$ to minimize the largest distance between any sample in $\mathcal{D}\backslash\mathcal{S}$ and its closest sample in $\mathcal{S}$;

**(4) Contextual Diversity (CD)** (Agarwal et al., 2020): Selects samples with diverse contexts based on predicted probabilities;

**(5) Margin** (Coleman et al., 2019): Selects samples with the smallest predictive probability difference between the most and second most likely labels;

**(6) Forgetting** (Toneva et al., 2018): Selects samples with more forgetting events (where a sample changes from correctly to incorrectly predicted more times), identifying them as hard samples;

**(7) GraNd** (Paul et al., 2021): Selects samples according to their GraNd scores, which quantify their contribution to the decline of training loss during early epochs;

**(8) Cal** (Margatina et al., 2021): Selects samples based on the KL divergence between the prediction of itself and that of its neighbors;

**(9) Glister** (Killamsetty et al., 2021b): Regards subset selection as a bilevel optimization problem and selects samples that satisfy optimization constraints;

**(10) Graph Cut(GC)** (Iyer et al., 2021): Greedily selects samples to maximize the Graph Cut function based on data gradients. Note that this implementation of GC uses gradients for data importance, unlike the version described in Section A.1, which uses features;

**(11) Moderate_DS (MDS)** (Xia et al., 2023): Selects samples with scores near the median of all samples, where scores represent distances to their class central feature;

**(12) MINimum distance (MIN)**: Selects samples closest to the central feature of their class.

### B.2. Performance of Baselines

In the main paper, we presented comparisons between the proposed method, RAM-APL, and several state-of-the-art methods on fine-grained datasets through illustrative analysis. Here, we provide the exact numerical results for reference and verification in Table 8, Table 9, and Table 10. To evaluate the overall effectiveness of each method across all sampling rates, we introduce a metric that computes the average performance improvement of each method relative to the baseline Random method, where Random is the most basic subset selection method. The results clearly demonstrate that RAM-APL achieves state-of-the-art performance, de-

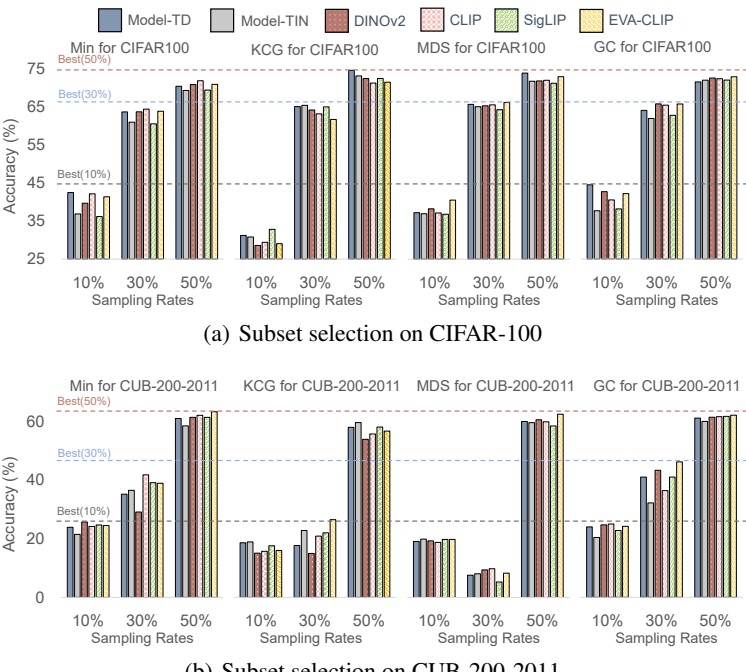

(a) Subset selection on CIFAR-100

(b) Subset selection on CUB-200-2011

*Figure 7.* Single-model study on the clean, coarse-grained CIFAR-100 dataset and the clean, fine-grained CUB-200-2011 dataset, which reinforces the insight that FMs are well-suited for (clean) fine-grained image datasets. Best viewed in color.

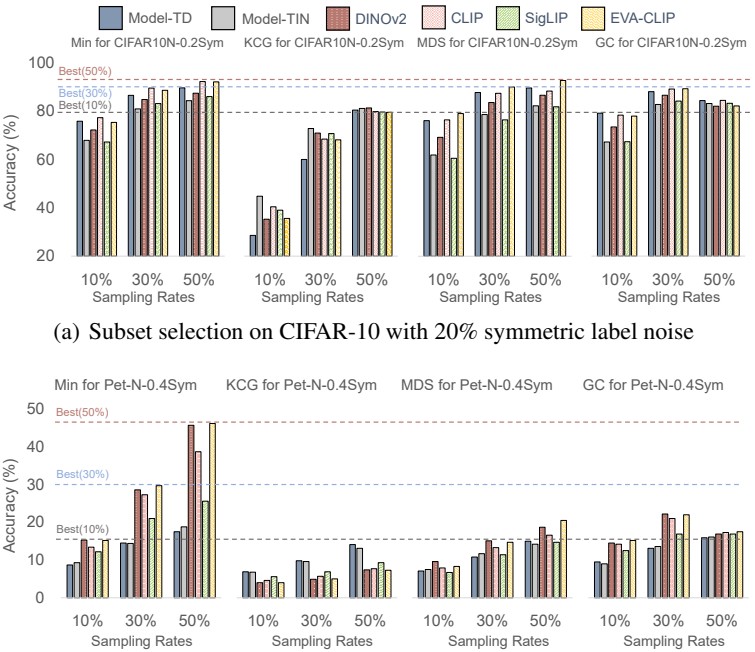

(a) Subset selection on CIFAR-10 with 20% symmetric label noise

(b) Subset selection on Oxford-IIIT Pet with 40% symmetric label noise

*Figure 8.* Single-model study on the coarse-grained CIFAR-10 dataset with label noise and the fine-grained Oxford-IIIT Pet dataset with label noise, which reinforces the insight that FMs are well-suited for (noisy) fine-grained image datasets. Best viewed in color.

livering the highest average improvement over the Random method at all sampling rates.

*Table 6.* Cross-architecture generalization of RAM-APL. The target model architecture is MobileNet-V3 (MBV3).

| Method | IE $\rightarrow$ Target Model | Sampling rates | | |
|---|---|---|---|---|
| | | 10% | 30% | 50% |
| Random | MBV3 $\rightarrow$ MBV3 | 10.9±1.1 | 42.1±3.6 | 61.6±1.9 |
| Forgetting | MBV3 $\rightarrow$ MBV3 | 13.3±0.8 | 42.0±2.0 | 61.0±2.3 |
| GC | MBV3 $\rightarrow$ MBV3 | 12.4±1.7 | 40.4±0.2 | 61.3±1.5 |
| MDS | MBV3 $\rightarrow$ MBV3 | 11.9±0.7 | 39.8±2.0 | 62.1±3.3 |
| MIN | MBV3 $\rightarrow$ MBV3 | 11.9±1.8 | 38.4±0.8 | 61.0±1.4 |
| **RAM-APL (Ours)** | (CLIP+DINOv2)$\rightarrow$ MBV3 | **13.6±0.3** | **45.7±0.9** | **62.3±1.4** |

*Table 7.* Average cosine distance of data pairs in the 70% subset of Pet.

| Method | IE | Class | | | | | Whole |
|---|---|---|---|---|---|---|---|
| | | 0 | 1 | 2 | 3 | 4 | |
| MIN | CLIP | 0.1617 | 0.1795 | 0.1176 | 0.1509 | 0.1327 | 0.2680 |
| RAM | CLIP+DINOv2 | 0.1695 | 0.1919 | 0.1259 | 0.1611 | 0.1392 | 0.2767 |
| RAM-APL | CLIP+DINOv2 | 0.1659 | 0.1986 | 0.1317 | 0.1597 | 0.1399 | 0.2787 |

## B.3. Cross-architecture Generalization

To evaluate whether our selected subsets remain effective across model architectures beyond ResNet, we conducted additional experiments on the Pet dataset using MobileNet-V3 as the target model. The results, presented in the Table 6, compare RAM-APL against five strong baselines that maintain identical architectures for their IEs and target models. RAM-APL consistently outperforms all baselines across different sampling rates, indicating its strong cross-architecture generalization ability.

## B.4. Visual Analysis of RAM Metric

As shown in Figure 9, we visualized samples from the Pet and Food101 datasets, arranging images within the same class from left to right based on their RAnking Mean (RAM) values, sorted from smallest to largest. The visualization reveals that images with smaller RAM values tend to exhibit clearer target objects and fewer background distractions. This indicates that RAM effectively distinguishes between easy and hard samples, highlighting its utility in subset selection.

## B.5. Influence of RAM and APL

To evaluate the influence of RAM and APL on subset diversity, we quantified the average cosine distance between data pairs. The results, summarized in Table 7, show that both RAM and RAM-APL yield subsets with greater average cosine distance compared to the MIN baseline, indicating an increase in both intra-class and overall diversity. For brevity, the table presents the results for first 5 classes out of 37 classes from the Pet dataset, along with the metric for the whole subset.

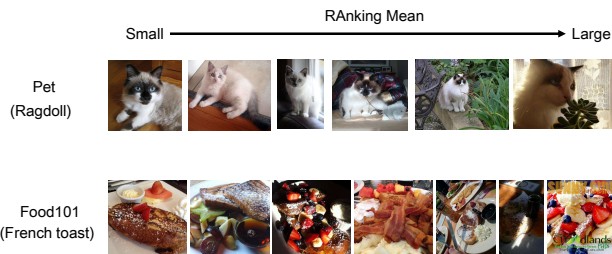

*Figure 9.* Visualisation of samples with RAM metric.

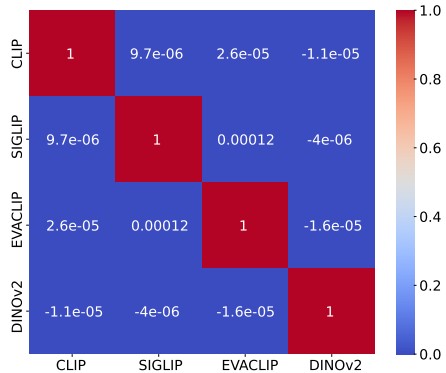

*Figure 10.* Cosine similarity matrix.

## B.6. Relationships Between Features Extracted by Different FMs

To explore the relationships between features extracted by different foundation models (FMs), we employed the cosine similarity metric. Specifically, we extracted features for the Pet dataset using DINOv2-VITs14, CLIP-VITl14, SigLIP-base-patch16-22, and EVA-CLIP-8B. To ensure uniformity, we first determined the minimum dimension $d$ among the feature matrices and applied the PCA algorithm (Pearson, 1901) to reduce all feature matrices to this dimension. Next, we computed the average cosine similarity between pairs of feature matrices and visualized the results in a heat map, as shown in Figure 10. The heat map reveals that the cosine similarity between features extracted by different foundation models is close to 0. This indicates that the feature distributions are largely independent in high-dimensional space, showing diverse data representation by the models.

*Table 8.* Comparison with baselines on the Pet dataset. "IE" denotes the Information Extractor. "Traditional" stands for models pre-trained on the Pet dataset for ten epochs. The means and variances are reported across five independent experiments with different random seeds. The best performance values are highlighted in **bold**, while the second-best results are marked in blue.

| Methods | IE | Sampling rates | | | | | Average Improvement over Random |
|---|---|---|---|---|---|---|---|
| | | 1% | 10% | 30% | 50% | 70% | |
| Random | - | 5.4±0.6 | 12.2±0.7 | 26.4±1.7 | 42.6±3.0 | 55.0±2.9 | - |
| Herding | | 4.7±0.5 | 10.8±0.7 | 22.5±2.7 | 38.2±1.7 | 53.2±2.7 | -2.44 |
| KCG | | 5.4±0.6 | 10.8±0.9 | 24.1±3.0 | 43.0±2.9 | 56.7±2.2 | -0.32 |
| CD | | 5.4±0.6 | 10.2±0.9 | 23.4±3.3 | 43.5±2.0 | 57.5±2.5 | -0.32 |
| Margin | | 5.0±0.3 | 11.6±0.4 | 25.3±1.5 | 42.0±2.6 | 54.8±1.5 | -0.58 |
| Forgetting | | 6.5±0.7 | 14.0±1.6 | 26.7±1.5 | 41.9±0.9 | 54.1±3.0 | +0.32 |
| GraNd | Traditional | 4.9±0.7 | 11.7±0.6 | 28.0±3.6 | 45.1±4.7 | 58.6±3.1 | +1.34 |
| Cal | | 5.9±0.6 | 13.9±0.8 | 27.1±1.4 | 41.5±1.2 | 54.3±3.7 | +0.22 |
| Glister | | 5.1±0.7 | 12.2±0.4 | 26.7±1.4 | 42.7±0.9 | 56.7±2.6 | +0.36 |
| GC | | 5.4±0.3 | 14.3±0.6 | 28.6±1.8 | 43.6±2.0 | 57.3±2.4 | +1.52 |
| MDS | | 4.7±0.4 | 11.9±1.6 | 24.5±2.3 | 42.0±2.2 | 55.4±3.4 | -0.62 |
| MIN | | 5.6±0.7 | 14.6±0.5 | 26.4±1.6 | 40.3±2.6 | 55.2±2.7 | +0.10 |
| Ours | CLIP+DINOv2 | **6.5±0.4** | **15.2±1.2** | **32.4±2.9** | **47.5±1.9** | **58.7±2.2** | **+3.74** |

*Table 9.* Comparison with baselines on the Food-101 dataset. "Traditional" stands for models pre-trained on the Food-101 dataset for ten epochs. The means and variances are reported across three independent experiments with different random seeds. The best performance values are highlighted in **bold**, while the second-best results are marked in blue.

| Methods | IE | Sampling rates | | | | | Average Improvement over Random |
|---|---|---|---|---|---|---|---|
| | | 1% | 10% | 30% | 50% | 70% | |
| Random | - | 6.8±0.4 | 45.5±0.5 | 69.5±0.2 | 76.2±0.3 | 79.6±0.3 | - |
| Herding | | 6.6±0.5 | 42.6±1.2 | 64.6±0.2 | 74.3±0.2 | 78.7±0.3 | -2.16 |
| KCG | | 4.7±0.3 | 35.0±0.7 | 67.2±0.2 | 75.9±0.2 | 79.6±0.2 | -3.04 |
| CD | | 3.9±0.1 | 27.7±0.6 | 66.2±0.4 | 75.7±0.2 | 80.0±0.1 | -4.82 |
| Margin | | 5.0±0.7 | 36.8±2.7 | 68.4±0.3 | 76.4±0.1 | 79.8±0.2 | -2.24 |
| Forgetting | | 9.5±0.5 | 53.4±0.5 | 71.7±0.1 | 77.3±0.1 | 79.9±0.1 | +2.84 |
| GraNd | Traditional | 2.9±0.2 | 18.4±0.4 | 59.7±0.5 | 74.7±0.2 | 80.0±0.1 | -8.38 |
| Cal | | 10.3±0.4 | 51.8±0.3 | 69.5±0.5 | 76.0±0.1 | 79.5±0.2 | +1.90 |
| Glister | | 6.9±0.3 | 38.8±1.1 | 67.4±0.4 | 75.9±0.4 | 80.1±0.2 | -1.70 |
| GC | | 10.8±0.4 | 54.4±0.3 | 71.3±0.4 | 76.8±0.2 | 79.5±0.3 | +3.04 |
| MDS | | 6.1±0.5 | 45.1±0.4 | 69.7±0.3 | 76.2±0.2 | 79.6±0.1 | -0.18 |
| MIN | | 9.4±0.2 | 52.1±0.6 | 70.7±0.7 | 76.7±0.1 | 79.8±0.1 | +2.22 |
| Ours | CLIP+DINOv2 | **12.3±0.3** | **56.1±0.6** | **72.3±0.1** | **77.9±0.1** | **81.2±0.2** | **+4.44** |

*Table 10.* Comparison with baselines on the CUB-200-2011 dataset. "Traditional" stands for models pre-trained on the CUB-200-2011 dataset for ten epochs. The means and variances are reported across three independent experiments with different random seeds. The best performance values are highlighted in **bold**, while the second-best results are marked in blue.

| Methods | IE | Sampling rates | | | | Average Improvement over Random |
|---------|-----|------|------|------|------|---|
| | | 10% | 30% | 50% | 70% | |
| Random | - | 21.6±0.9 | 27.5±5.4 | 59.5±0.3 | 67.0±0.5 | - |
| Herding | | 14.3±1.6 | 17.5±6.0 | 55.1±0.7 | 65.7±0.1 | -5.75 |
| KCG | | 18.6±2.6 | 17.7±4.3 | 58.0±0.9 | 66.2±0.3 | -3.78 |
| CD | | 16.4±1.0 | 26.0±8.3 | 56.7±0.7 | 66.2±0.2 | -2.58 |
| Margin | | 16.0±1.0 | 19.6±4.8 | 56.5±0.4 | 66.2±0.7 | -4.33 |
| Forgetting | | 17.3±0.7 | 22.7±3.1 | 59.1±0.9 | 68.1±0.2 | -2.10 |
| GraNd | Traditional | 13.2±1.3 | 20.6±9.1 | 55.0±0.6 | 66.2±0.8 | -5.15 |
| Cal | | 22.0±1.7 | 32.7±4.5 | 60.7±0.4 | 68.0±0.4 | +1.95 |
| Glister | | 15.5±0.4 | 21.8±4.7 | 56.6±0.7 | 65.9±0.5 | -3.95 |
| GC | | 22.4±0.1 | 36.1±10.4 | 61.1±0.7 | 67.1±0.2 | +2.78 |
| MDS | | 19.1±0.4 | 7.6±2.9 | 60.0±0.7 | 67.3±0.5 | -5.40 |
| MIN | | 24.1±0.5 | 32.8±9.4 | 61.0±0.7 | 67.7±0.1 | +2.50 |
| Ours | CLIP+DINOv2 | **25.1±0.4** | **42.6±3.1** | **63.5±0.3** | **70.0±0.3** | **+6.40** |

