# OpenReview forum: "Foundation Model Insights and a Multi-Model Approach for Superior Fine-Grained One-shot Subset Selection"
_ICML.cc/2025/Conference — ICML 2025 oral_

### Official Review · Reviewer_g7Wk · 2025-03-09

**Overall Recommendation:** 4

**Summary:**

This paper explores the use of foundation models (FMs) for one-shot subset selection, focusing on fine-grained image datasets. The authors find that FMs outperform traditional information extractors (IEs) in fine-grained tasks but struggle with noisy, coarse-grained datasets. To address this, they propose RAM-APL, a multi-FM framework that combines intra-class (via Ranking Mean, RAM) and inter-class (via Accuracy of Pseudo-class Labels, APL) feature analysis for improved subset selection. The method is evaluated on fine-grained datasets like Oxford-IIIT Pet and CUB-200-2011, demonstrating its effectiveness.

**Claims And Evidence:**

The authors demonstrate RAM-APL's superiority on fine-grained datasets like Oxford-IIIT Pet and CUB-200-2011, with ablation studies validating its components. However, the claim that FMs underperform on noisy, coarse-grained datasets could benefit from broader experiments. Overall, the evidence is solid but could be further reinforced.

**Essential References Not Discussed:**

From my perspective, this article does not omit any essential references.

**Experimental Designs Or Analyses:**

The experimental design is well-structured, with appropriate benchmark datasets (e.g., Oxford-IIIT Pet, CUB-200-2011) and a comprehensive set of baseline methods for comparison.  The inclusion of ablation studies and parameter analyses is good, as they provide valuable insights into the contributions of different components of the proposed RAM-APL method.  However, the ablation experiments in Tables 1, 2, and 3 are limited to sampling rates of 1%, 50%, and 70%, which raises some concerns.  The 1% sampling rate seems unconventional and may not reflect practical scenarios, while the 70% rate sometimes underperforms, as seen in the results.  A more balanced evaluation, including the 10% and 30% sampling rates used in Figure 3, would provide a clearer understanding of the method's performance across a wider range of realistic settings.

**Methods And Evaluation Criteria:**

The proposed RAM-APL method addresses fine-grained subset selection by leveraging multiple foundation models and combining intra-class and inter-class feature analysis, which appears reasonable for the task.  The evaluation criteria, including accuracy on datasets like Oxford-IIIT Pet and CUB-200-2011, are appropriate for assessing performance.  However, the evaluation could be expanded to include more diverse datasets, particularly those with noisy or coarse-grained characteristics, to better validate the method's robustness and generalizability.  Overall, the methods and evaluation criteria are suitable, though broader validation could strengthen the findings.

**Other Comments Or Suggestions:**

N/A

**Other Strengths And Weaknesses:**

There are no other strengths and weaknesses. Here I summarize the strengths and weaknesses I answered in the previous parts.

Strengths:

1. Clarity of Methodology: The paper presents the proposed method, RAM-APL, in a straightforward and concise manner, making it easy to understand and follow.
2. Comprehensive Experimental Setup: The experiments are well-designed and cover a range of datasets and scenarios, providing a thorough evaluation of the method's effectiveness.

Weaknesses:

1. Lack of Analysis on FM Performance in Coarse-Grained Tasks: While the paper highlights that foundation models (FMs) underperform in coarse-grained, noisy datasets, it does not provide a detailed analysis or explanation for this behavior. Additional experiments or theoretical insights could help clarify why FMs struggle in these scenarios.
2. Limited Sampling Rate Evaluation in Ablation Studies: Tables 1, 2, and 3 only compare results at 1%, 50%, and 70% sampling rates. The 1% rate is unconventional and may not reflect practical use cases, while the 70% rate sometimes underperforms. It would be more informative to include results for 10% and 30% sampling rates, as shown in Figure 3, to provide a more balanced and realistic evaluation of the method's performance.

**Questions For Authors:**

Please see the weaknesses of the ``Other Strengths And Weaknesses'' part, where there is a summary.

**Relation To Broader Scientific Literature:**

The key contributions of this paper are closely related to the broader literature on subset selection and foundation models.

**Theoretical Claims:**

The paper does not present any explicit theoretical claims or proofs, which focuses on the empirical evaluation of the proposed RAM-APL method.

---

> ### Author Rebuttal · Authors · 2025-03-31
>
> Thank you for your insightful feedbacks! We address your questions in the following responses.
>
> ___
>
> **W1: A detailed analysis of why FM as IE underperform on coarse-grained datasets with noisy labels.**
>
> A1: We sincerely appreciate the reviewer's insightful comments. Due to the character limit in the rebuttal, we kindly refer the reviewer to our response to Reviewer aGx6's "W1&Q1: A deeper discussion of why FM as IE performs poorly on coarse-grained datasets with noisy labels."
>
> ___
>
> **W2: Limited sampling rate evaluation in ablation studies.**
>
> A2: We sincerely appreciate the reviewer’s insightful feedback and acknowledge the importance of conducting ablation studies across diverse sampling rates.
>
> To address this concerns, we have conducted additional Ablation Studies at 10% and 30% sampling rates. The revised Tables 1, 2, and 3 are provided below.
>
> Table 1. Ablation study based on Pet.
> | Method    | Information Extractor (IE) | 1%       | 10%      | 30%       | 50%       | 70%       |
> |-----------|---------------------------|----------|----------|-----------|-----------|-----------|
> | MIN |Model-TD| 5.6±0.7  | 14.6±0.5 | 26.4±1.6  | 40.3±2.6  | 55.2±2.7  |
> | MIN | CLIP | 5.6±0.2  | 15.4±1.0 | 29.3±2.4  | 45.9±1.8  | 56.3±0.7  |
> |  MIN | DINOv2 | 6.2±0.1 | **15.5±0.7** | 32.0±1.4 | 46.8±2.0 | **60.5±2.9** |
> | RAM | CLIP+DINOv2 | 5.9±0.3  | 15.1±0.5 | **33.1±2.3**  | 47.1±1.4  | 56.5±2.7  |
> | RAM-APL | CLIP+DINOv2 | **6.5±0.4**  | 15.2±1.2 | 32.4±2.9  | **47.5±1.9**  | 58.7±2.2  |
>
> Across various sampling rates, RAM consistently outperforms MIN (CLIP as IE), while RAM-APL further improves performance, reaching levels comparable to DINOv2. Though RAM-APL (CLIP+DINOv2) demonstrates overall superior performance, its effectiveness at 70% sampling can be improved. In future work, we aim to enhance our method’s effectiveness at high sampling rates to further improve its practical utility.
>
> Table 2. Comparison of the performance of our method using different numbers of foundation models as information extractors.
> | DINOv2 | CLIP | SigLIP | EVA-CLIP | 1%       | 10%      | 30%       | 50%       | 70%       |
> |----------|------------|------------|-----------|----------|----------|-----------|-----------|-----------|
> | ✓        |            |            |           | 5.9±0.3  | 15.4±1.1 | 31.6±2.3  | 47.7±1.1  | 57.9±4.1  |
> |          | ✓          |            |           | 5.7±0.4  | 15.0±0.2 | 27.9±1.2  | 43.6±1.9  | 57.0±0.4  |
> |          |            | ✓          |           | 6.6±0.3  | 14.1±1.0 | 28.8±1.1  | 43.9±1.7  | 55.1±2.6  |
> |          |            |            | ✓         | 5.4±0.3  | 15.0±0.6 | 30.2±2.5  | 44.4±2.3  | 56.6±1.8  |
> | ✓        | ✓          |            |           | 6.5±0.4  | 15.2±1.2 | 32.4±2.9  | 47.5±1.9  | **58.7±2.2**  |
> | ✓        |            | ✓          |           | 5.9±0.3  | 16.2±0.1 | 31.4±3.2  | 45.0±1.3  | 58.6±1.2  |
> | ✓        |            |            | ✓         | 6.0±0.6  | 16.0±0.9 | **35.8±2.9**  | 46.5±1.8  | 54.9±3.5  |
> |          | ✓          | ✓          |           | 6.4±0.2  | 15.1±0.4 | 29.8±1.6  | 45.9±1.3  | 56.2±2.7  |
> |          | ✓          |            | ✓         | 5.9±0.3  | 15.5±0.7 | 31.4±1.7  | 44.2±2.2  | 55.9±1.8  |
> |          |            | ✓          | ✓         | **6.7±0.4**  | 16.2±0.6 | 34.7±0.3  | 45.7±0.8  | 56.6±2.4  |
> | ✓        | ✓          | ✓          |           | 6.2±0.8  | 15.6±0.5 | 33.2±1.4  | **48.3±1.1**  | 57.6±0.1  |
> | ✓        | ✓          |            | ✓         | 6.0±0.4  | **17.5±1.0** | 35.2±1.8 | 47.9±1.5  | 55.6±2.1  |
> | ✓        |            | ✓          | ✓         | 6.1±0.3  | 16.8±0.6 | 34.4±2.1  | 47.0±2.0  | 55.1±1.6  |
> |          | ✓          | ✓          | ✓         | 6.1±0.2  | 16.1±0.3 | 33.9±1.4  | 46.8±1.5  | 55.1±0.5  |
> | ✓        | ✓          | ✓          | ✓         | 6.5±0.2  | 16.8±1.1 | 34.0±2.7  | 46.3±0.5  | 56.9±1.1  |
>
> We observe that leveraging multiple foundation models outperforms using a single model. The optimal balance of computational efficiency, memory usage, and performance is achieved with DINOv2 + CLIP. For the highest overall accuracy, DINOv2+CLIP+EVA-CLIP is recommended. These findings validate the benefits of multi-model selection, and the results will be included in the supplementary material.
>
> Tabel 3. Comparison of feature fusion strategies.
> | Fusion Method  | 1%       | 10%      | 30%       | 50%       | 70%       |
> |----------------|----------|----------|-----------|-----------|-----------|
> | Concatenate    | 5.9±0.4  | 16.3±0.4 | 31.7±1.3  | 47.7±3.0  | 57.8±1.2  |
> | Ours| 6.5±0.4 | 15.2±1.2| 32.4±2.9 | 47.5±1.9 |58.7±2.2|
>
> We observe that Our strategy outperforms Concatenate, especially at higher sampling rates, which are crucial for practical applications. To maximize the performance of multi-model method at high sampling rates, we adopt the Ours fusion strategy. The findings and new results will be included in the supplementary material.

---

### Official Review · Reviewer_35yt · 2025-03-11

**Overall Recommendation:** 4

**Summary:**

The paper makes comparisons between traditional information extractors (IEs) and a single foundation model (FM) on a series of datasets to explore scenarios in which a single FM would be advantageous as an IE. It reveals that a single FM performs poorly on coarse-grained image datasets with noisy labels and performs well on fine-grained image datasets with clean and noisy labels. The paper introduces a one-shot subset selection approach (called RAM-APL) tailed for fine-grained datasets, which ingeniously maps the misaligned features extracted by an ensemble of FMs into a unified distance ranking space, considering both intra-class distribution and inter-class distribution of samples. Experiments demonstrate the SOTA selection performance of RAM-APL on three fine-grained image datasets.

**Claims And Evidence:**

The claims regarding the foundation model insights and the effectiveness of RAM-APL in the paper are supported by strong empirical evidence. However, additional analysis on fine-grained datasets with noisy labels could further strengthen the claims.

**Essential References Not Discussed:**

N/A

**Experimental Designs Or Analyses:**

To explore scenarios in which a single FM would be advantageous as an IE, the paper employs systematic and rigorous experiments to analyze the impact of various factors (such as coarse-grained and fine-grained, labels that are clean or noisy, and balanced or unbalanced class distributions) and provides in-depth discussions of the results. Besides, the experimental design is robust in validating the effectiveness of RAM-APL, and the chosen datasets (Oxford-IIIT Pet, Food-101, and CUB-200-2011) are well suited for evaluating subset selection methods in the context of fine-grained image classification. The paper compares RAM-APL against multiple baseline methods, showing clear improvements.

However, it would be beneficial to include additional experiments to explicitly examine the performance of subset selection methods on fine-grained datasets with noisy labels. The paper highlights the strengths of FMs as IEs on both fine-grained datasets with clean and noisy labels, so more explicit comparisons between RAM-APL and other methods on fine-grained datasets with noisy labels would further strengthen the effectiveness of RAM-APL.

**Methods And Evaluation Criteria:**

The proposed approach is reasonable, but the evaluation is limited to fine-grained datasets. A broader range of datasets would help confirm RAM-APL’s robustness.

**Other Comments Or Suggestions:**

- Consider including a code release or a link to a public repository for reproducibility purposes.

**Other Strengths And Weaknesses:**

Strengths
- The paper is well-written and clear and presents a novel contribution to subset selection in fine-grained image classification.
- The paper systematically compares FMs and traditional IEs through rigorous experiments, offering practical insights into their relative performance across diverse image datasets.
- The paper is highly original in combining multiple FMs for subset selection, a novel approach that significantly improves performance in fine-grained datasets.
- The paper provides comprehensive empirical results, strengthening its practical impact.

Weaknesses
1. The paper provides convincing results on three classical image fine-grained datasets. However, the paper does not compare the performance of RAM-APL with other methods on fine-grained image datasets with noisy labels. Experiments on fine-grained image datasets with noisy labels are important to further demonstrate the effectiveness of RAM-APL.
2. The paper lacks a deeper discussion of why FM as IE performs well on fine-grained datasets with noisy labels and poorly on coarse-grained datasets with noisy labels. For example, which specific types of images or classes does FM as IE perform better on, and which classes does it not perform well on? Understanding these nuances would help in understanding the advantages of FM as an IE and would help in adapting RAM-APL to other domains.

**Questions For Authors:**

Please see [Weaknesses] 1-2 in the Other Strengths And Weaknesses. If authors address them, reviewer would like to change the rating.

**Relation To Broader Scientific Literature:**

The paper effectively situates itself within the subset selection literature, particularly focusing on feature-based subset selection. The use of FMs for subset selection is both relevant and timely, especially as research increasingly relies on large pre-trained models.

**Theoretical Claims:**

The paper does not present formal theoretical proof, as it is largely focused on empirical evaluation. The conceptual framework of the RAM-APL method is well-explained, and the reliance on empirical analysis is justified. No significant issues were found in the presentation of the algorithmic ideas.

---

> ### Author Rebuttal · Authors · 2025-03-31
>
> Thank you for your positive feedbacks! We address your questions in the following responses.
>
> ___
>
> **W1: Evaluation on fine-grained image datasets with noisy labels.**
>
> A1: We sincerely appreciate the reviewer’s insightful suggestion. We acknowledge the importance of evaluating the effectiveness of our approach with other selection methods on fine-grained image datasets with noisy labels.
>
> To address this concerns, we conducted additional experiments (as detailed in the tables below) on the Oxford-IIIT Pets dataset with 20% symmetric label noise and with 40% symmetric label noise. Subsets are sampled following the same experimental setup described in the manuscript.
>
> Dataset: Oxford-IIIT Pets dataset with 20% symmetric label noise
> | Method     | IE | 1%       | 10%      | 30%       | 50%       | 70%       | 100%     |
> |------------|---------------|----------|----------|-----------|-----------|-----------|----------|
> | Random     | -             | 4.9±0.7  | 10.0±1.0 | 16.7±0.6  | 25.3±0.4  | 33.4±2.6  | 42.7±1.8 |
> | Harding    | Model-TD   | 5.0±0.1  | 8.1±1.8  | 15.3±1.8  | 20.7±0.5  | 33.1±0.4  | 42.7±1.8 |
> | KCG        | Model-TD   | 5.3±1.4  | 7.8±0.8  | 15.4±1.1  | 22.3±1.7  | 32.2±1.5  | 42.7±1.8 |
> | CD         | Model-TD   | 5.2±0.4  | 6.6±0.8  | 13.7±0.2  | 22.4±1.6  | 32.1±1.4  | 42.7±1.8 |
> | Margin     | Model-TD   | 4.7±0.1  | 9.0±0.7  | 16.3±0.5  | 23.9±0.6  | 33.5±1.2  | 42.7±1.8 |
> | Forgetting | Model-TD   | 5.9±0.6  | 11.5±0.9 | 18.7±1.3  | 29.5±0.9  | 36.9±0.4  | 42.7±1.8 |
> | GraNd      | Model-TD   | 4.3±0.2  | 7.8±0.8  | 15.6±0.7  | 22.9±1.5  | 32.4±2.1  | 42.7±1.8 |
> | Cal        | Model-TD   | 6.2±0.8  | 12.2±0.6 | 22.2±2.6  | 29.4±1.3  | 38.7±0.8  | 42.7±1.8 |
> | Glister    | Model-TD   | 4.8±0.2  | 10.5±1.1 | 17.2±1.0  | 26.1±2.6  | 34.7±2.2  | 42.7±1.8 |
> | GC         | Model-TD   | 5.2±0.6  | 12.8±1.5 | 20.3±1.3  | 27.0±0.6  | 32.9±0.9  | 42.7±1.8 |
> | MDS        | Model-TD   | 3.8±0.4  | 9.8±0.3  | 17.1±0.6  | 24.3±1.7  | 30.7±3.1  | 42.7±1.8 |
> | MIN        | Model-TD   | 5.6±0.2  | 11.6±0.4 | 19.8±1.4  | 28.0±2.2  | 35.3±2.5  | 42.7±1.8 |
> | **Ours**   | **CLIP+DINOv2** | **6.7±0.3** | **16.7±0.3** | **32.5±1.8** | **46.0±1.6** | **56.7±0.7** | 42.7±1.8 |
>
> Dataset: Oxford-IIIT Pets dataset with 40% symmetric label noise
> | Method     | IE | 1%       | 10%      | 30%       | 50%       | 70%       | 100%     |
> |------------|---------------|----------|----------|-----------|-----------|-----------|----------|
> | Random     | - | 5.1±0.5  | 8.0±0.6  | 12.6±0.6  | 15.0±0.3  | 19.1±0.5  | 23.0±0.6 |
> | Harding    | Model-TD | 4.4±0.2  | 6.3±0.5  | 11.1±0.9  | 13.1±0.6  | 18.2±1.3  | 23.0±0.6 |
> | KCG        | Model-TD | 4.9±0.8  | 6.3±0.5  | 9.9±1.2   | 14.3±1.1  | 18.1±0.9  | 23.0±0.6 |
> | CD         | Model-TD | 4.8±0.8  | 6.3±0.5  | 10.3±0.8  | 14.0±0.3  | 17.7±1.2  | 23.0±0.6 |
> | Margin     | Model-TD | 4.1±0.3  | 7.0±0.8  | 11.1±0.9  | 14.3±0.9  | 19.0±0.8  | 23.0±0.6 |
> | Forgetting | Model-TD | 5.4±0.8  | 10.2±1.6 | 12.9±0.8  | 17.2±0.4  | 21.4±0.9  | 23.0±0.6 |
> | GraNd      | Model-TD | 4.4±0.9  | 6.7±1.0  | 10.2±0.5  | 14.5±1.6  | 18.8±1.2  | 23.0±0.6 |
> | Cal        | Model-TD | 5.4±0.3  | 10.6±1.1 | 14.9±1.1  | 18.9±1.0  | 22.2±1.2  | 23.0±0.6 |
> | Glister    | Model-TD | 5.2±0.3  | 7.6±1.1  | 12.4±0.8  | 18.3±0.8  | 21.8±1.6  | 23.0±0.6 |
> | GC         | Model-TD | 4.9±0.7  | 9.7±1.1  | 12.8±0.7  | 15.4±0.8  | 20.5±1.7  | 23.0±0.6 |
> | MDS        | Model-TD | 3.9±0.2  | 7.2±0.3  | 12.0±0.2  | 15.0±1.5  | 18.5±0.8  | 23.0±0.6 |
> | MIN        | Model-TD | 5.3±0.4  | 9.4±0.7  | 14.3±0.7  | 18.3±0.6  | 20.9±0.6  | 23.0±0.6 |
> | **Ours**   | **CLIP+DINOv2** | **6.1±0.3** | **15.0±1.2** | **30.4±1.7** | **44.8±0.1** | **42.6±0.8** | 23.0±0.6 |
>
> ("IE" means information extractor, "Model-TD" denotes the model trained on the full set for 10 epochs.)
>
> We observe that RAM-APL consistently outperforms all baselines across different sampling rates on each noisy fine-grained  dataset, demonstrating its effectiveness.
>
> Your suggestion has been highly valuable. Through experimental analysis, we have identified the significant advantages of designing selection algorithms based on foundation models for noisy datasets, which motivates us to explore more effective foundation model-based denoising approaches in future work. The above experimental results and discussions will be included in the supplementary material.
>
> ___
>
> **W2: A deeper discussion of why FM as IE performs poorly on coarse-grained datasets with noisy labels.**
>
> A2: We sincerely appreciate the reviewer's insightful comments. Due to the character limit in the rebuttal, we kindly refer the reviewer to our response to Reviewer aGx6's "W1&Q1: A deeper discussion of why FM as IE performs poorly on coarse-grained datasets with noisy labels."
>
> ___
>
> **S1: Code release.**
>
> A3: We thank the reviewer for this suggestion. We will release the full implementation code in a public repository upon paper acceptance to ensure reproducibility.

---

> > ### Comment · Reviewer_35yt · 2025-04-07
> >
> > Thanks to the authors for their positive response and detailed rebuttal. The authors have addressed my concerns.

---

### Official Review · Reviewer_g2WV · 2025-03-11

**Overall Recommendation:** 4

**Summary:**

To investigate whether foundation models (FMs) can truly replace task-specific information extractors (IEs) in subset selection, this paper examines the effectiveness of FMs as IEs for one-shot subset selection. Through extensive experiments across a set of image datasets, this paper identifies the strengths and limitations of FMs as IEs: they excel on fine-grained image datasets but underperform on coarse-grained datasets with noisy labels. To capitalize on the complementary strengths of multiple FMs and overcome limitations in existing feature-based selection methods, this paper introduces RAM-APL, which maps misaligned features from multiple FMs into a unified distance ranking space, considering intra-class and inter-class distributions. The selection methods are evaluated on three fine-grained classification datasets.

**Claims And Evidence:**

Yes. The claims in this paper are generally supported by clear experimental results, particularly in demonstrating that RAM-APL improves subset selection on fine-grained image datasets.

**Essential References Not Discussed:**

No

**Experimental Designs Or Analyses:**

Yes.  The experimental design is rigorous and well-structured:
1. The evaluation considers three fine-grained image datasets (CUB-200-2011, Oxford-IIIT Pets, Food-101), making the conclusions well-supported in the targeted domain.
2. A range of baselines is compared, including random selection, single-FM approaches (DINOv2, CLIP, et al.), and other subset selection methods.
3. The ablation study analyzes hyperparameters (\alpha, \beta) and the effect of different FM combinations, showing that DINOv2 + CLIP provides the best results.

However, a few concerns:
1. This paper does not assess whether the selected subsets generalize across different model architectures. A key question is whether subsets selected by RAM-APL would maintain their effectiveness when applied to architectures beyond ResNet.
2. This paper claims FMs are ineffective on coarse-grained datasets with noisy labels but does not analyze why in depth. A more detailed study (e.g., feature visualizations or error analysis) would help substantiate this finding.

**Methods And Evaluation Criteria:**

Yes. The methodology is well-motivated for subset selection, and the benchmark datasets are appropriate.

**Other Comments Or Suggestions:**

See the Weakness part.

**Other Strengths And Weaknesses:**

Strengths:
1. This paper structure is reasonable, with each component of the proposed method clearly explained, making it easy to understand and implement.
2. This paper introduces a well-motivated and innovative approach to subset selection. RAM-APL effectively harnesses the complementary advantages of multiple foundation models, addressing the variability in FM performance across datasets and selection methods. The empirical evaluations are thorough and provide strong evidence supporting the effectiveness of the proposed method.

Weaknesses:
1.This paper does not assess whether the selected subsets generalize across different model architectures. Would the subsets selected by RAM-APL retain their effectiveness when applied to architectures beyond ResNet?
2. While this paper argues that FMs perform poorly on coarse-grained datasets with noisy labels, it lacks an in-depth analysis of the underlying reasons. Incorporating feature visualizations or error analysis could provide stronger empirical justification for this claim.
3. The analysis is somewhat limited, primarily focusing on accuracy. Additional insights, such as diversity or difficulty analysis of the selected subsets, would enhance the evaluation.

**Questions For Authors:**

1. Would the subsets selected by RAM-APL retain their effectiveness when applied to architectures beyond ResNet?
2.Why do FMs struggle with coarse-grained datasets containing noisy labels but perform well on fine-grained image datasets?
3. How do RAM and APL strategies influence the distribution of representations in selected subsets?

**Relation To Broader Scientific Literature:**

This paper relates to subset selection approaches but differs by leveraging multiple FMs to form a unified ranking space.

**Theoretical Claims:**

Yes. Since this paper is largely data-driven, there are no formal proofs for the theoretical claims, but the empirical justification is sound.

---

> ### Author Rebuttal · Authors · 2025-03-31
>
> Thank you for your positive feedbacks! We address your questions in the following responses.
>
> ___
>
> **W1&Q1: Cross-architecture generalization of RAM-APL.**
>
> A1: We sincerely appreciate the reviewer’s insightful question regarding the cross-architecture generalization of RAM-APL. We acknowledge the importance of evaluating whether our selected subsets remain effective across different model architectures beyond ResNet.
>
> To address this concerns, we conducted additional experiments on the Oxford-IIIT Pets dataset (Pets) using MobileNet-V3 as the target model. The results, presented in the table below, compare RAM-APL against five strong baselines that maintain identical architectures for their information extractors (IE) and target models.
>
> MobileNet-V3 (MBV3)
> | Method	| IE→Target Model | 10%	| 30% | 50% |
> |----------------------|--------------------------|-------------|-------------|-------------|
> | Random	| MBV3 → MBV3| 10.9±1.1 | 42.1±3.6 | 61.6±1.9 |
> | Forgetting	| MBV3 → MBV3| 13.3±0.8 | 42.0±2.0 | 61.0±2.3 |
> | GC		| MBV3 → MBV3| 12.4±1.7    | 40.4±0.2    | 61.3±1.5 |
> | MDS		| MBV3 → MBV3| 11.9±0.7    | 39.8±2.0    | 62.1±3.3 |
> | MIN		| MBV3 → MBV3| 11.9±1.8    | 38.4±0.8    | 61.0±1.4 |
> | **RAM-APL (Ours)** | (CLIP+DINOv2)→ MBV3 | **13.6±0.3** | **45.7±0.9** | **62.3±1.4** |
>
> We observe that RAM-APL consistently outperforms all baselines across different sampling rates, indicating its strong cross-architecture generalization ability.
>
> Your suggestion has been highly valuable, inspiring us to further explore multi-model subset selection in broader cross-architecture settings in future work. The above experimental results and discussion will be included in the supplementary material.
>
> ___
>
> **W2&Q2: A deeper discussion of why FMs struggle with coarse-grained datasets containing noisy labels but perform well on fine-grained image datasets.**
>
> A2: We sincerely appreciate the reviewer's insightful comments. Due to the character limit in the rebuttal, we kindly refer the reviewer to our response to Reviewer aGx6's "W1&Q1: A deeper discussion of why FM as IE performs poorly on coarse-grained datasets with noisy labels."
>
> ___
>
> **W3&Q3: How do RAM and APL influence the distribution of representations in selected subsets?**
>
> A3: We sincerely appreciate the reviewer’s insightful question regarding the influence of RAM and APL strategies on the distribution of representations in the selected subsets. We acknowledge the importance of analyzing how these strategies shape the feature space and their impact on sample diversity and representativeness.
>
> To address this concern, we conducted additional experiments and analyzed the feature distributions of different selection strategies. Specifically, we examined the average cosine distance between data pairs within the selected subsets, which provides insights into intra-class and overall diversity. The results are summarized in the table below:
>
> Table. Average cosine distance of data pairs in the subset
> | Method    | IE          | Class 0   | Class 1   | Class 2   | Class 3   | Class 4   | Whole subset |
> |-----------|-------------|-----------|-----------|-----------|-----------|-----------|--------------|
> | Min       | CLIP        | 0.1617    | 0.1795    | 0.1176    | 0.1509    | 0.1327    | 0.2680       |
> | RAM       | CLIP+DINOv2 | 0.1695    | 0.1919    | 0.1259    | 0.1611    | 0.1392    | 0.2767       |
> | RAM-APL   | CLIP+DINOv2 | 0.1659    | 0.1986    | 0.1317    | 0.1597    | 0.1399    | **0.2787**   |
>
> From these results, we observe that RAM and RAM-APL lead to a more diverse feature distribution in the selected subset compared to Min-based selection. The whole-subset average cosine distance is highest under RAM-APL (0.2787), indicating that it selects more diverse samples overall, improving coverage of the feature space. Moreover, the per-class distances suggest that RAM-APL encourages a balance between inter-class and intra-class diversity, with slightly higher values in harder-to-distinguish classes.
>
> Furthermore, the t-SNE visualizations in Figures 9-11  (https://anonymous.4open.science/r/RAM-APL-DED5/README.md) further confirm these findings. Compared to Min-based selection, which tends to concentrate samples within certain regions of the feature space, RAM and RAM-APL distribute samples more broadly across the space, ensuring better representational coverage. This suggests that our approach enhances model performance by capturing a more comprehensive representation of the dataset.
>
> Your suggestion has been highly valuable in strengthening our analysis. The above results and discussions will be included in the supplementary material to provide a clearer understanding of the impact of our proposed selection strategies.

---

> > ### Comment · Reviewer_g2WV · 2025-04-07
> >
> > After reading the response, the authors have addressed my concerns. Thus, I support accepting this paper.

---

### Official Review · Reviewer_aGx6 · 2025-03-12

**Overall Recommendation:** 4

**Summary:**

This paper investigates one-shot subset selection using Foundation Models (FMs) to reduce deep learning training costs by improving efficiency. Traditional Information Extractors (IEs) rely on models pre-trained on the target dataset, introducing dataset dependency. The paper addresses two key questions: (1) Can FM-based subset selection outperform traditional IE-based methods across diverse datasets? (2) Do all FMs perform equally well for subset selection? Experimental results show that FMs excel on fine-grained datasets but underperform on coarse-grained datasets with noisy labels. Based on these findings, the authors propose RAM-APL (RAnking Mean-Accuracy of Pseudo-class Labels), a novel method that leverages multiple FMs to enhance subset selection performance on fine-grained datasets. Extensive experiments validate the superiority of RAM-APL on three fine-grained datasets.

**Claims And Evidence:**

The main claims of the paper are supported by experimental data. For instance, the superiority of FMs as IEs on fine-grained datasets and the effectiveness of the RAM-APL method are validated through experiments. However, some conclusions (e.g., the limitations of FMs on coarse-grained datasets) lack deeper explanations.

**Essential References Not Discussed:**

The paper cites a wide range of related literature but omits some key works. For example, FM applications on noisy datasets (e.g., “CLIPCleaner: Cleaning Noisy Labels with CLIP” by Chen Feng et al., 2024) are highly relevant. The authors are encouraged to include relevant references and discuss the implications.

**Experimental Designs Or Analyses:**

The experimental design of Single-model Study is comprehensive, covering multiple datasets (e.g., CIFAR-10, CIFAR-10N, Oxford-IIIT Pet) and different FMs (e.g., DINOv2, CLIP). The results demonstrate that FMs perform well on fine-grained datasets but struggle on coarse-grained datasets with noisy labels. The RAM-APL method significantly improves performance on fine-grained datasets by combining the feature extraction capabilities of multiple FMs.

A limitation of the experimental design is the lack of in-depth analysis of why FMs underperform on coarse-grained datasets. For example, is this due to the feature distribution or noise levels in the datasets?

**Methods And Evaluation Criteria:**

The proposed RAM-APL method significantly improves subset selection performance on fine-grained datasets by leveraging the feature extraction capabilities of multiple FMs. The method is well-designed, and the evaluation criteria (e.g., prediction accuracy) are appropriate for the subset selection task. The experimental results demonstrate that RAM-APL outperforms SOTA methods on multiple datasets, validating its effectiveness.

**Other Comments Or Suggestions:**

NA

**Other Strengths And Weaknesses:**

Strengths:
1.	The findings of the effectiveness of FMs for subset selection are interesting.
2.	The proposed RAM-APL method significantly improves subset selection performance on fine-grained datasets.
3.	The experimental design is comprehensive, covering multiple datasets and FM combinations.
4.	The supplementary material provides detailed experimental explanations and results, enhancing the paper's credibility.

Weaknesses:
1.	The paper lacks a theoretical explanation for the underperformance of FMs on coarse-grained datasets with noisy labels.
2.	The paper lacks a theoretical analysis of the RAM-APL method, explaining why it effectively leverages the complementary strengths of multiple FMs.
3.	The discussion of related FM literature is insufficient.

**Questions For Authors:**

1.	Why do FMs underperform on coarse-grained datasets with noisy labels? Is this related to feature distribution or noise levels in the datasets?
2.	Can the authors provide a theoretical analysis of the RAM-APL method, explaining why it effectively leverages the complementary strengths of multiple FMs?
3.	Have the authors considered applying the RAM-APL method to other tasks, such as few-shot learning or various noisy datasets?

**Relation To Broader Scientific Literature:**

The paper clearly situates itself within the existing literature. Traditional subset selection methods rely on IEs pre-trained on the target dataset, which introduces dataset dependency. By introducing FMs, the paper proposes a dataset-agnostic subset selection method, expanding the scope of subset selection research. However, the paper does not sufficiently discuss the relationship with existing FM-related work, such as FM applications to few-shot learning or noisy datasets.

**Theoretical Claims:**

The paper does not provide rigorous theoretical proofs but validates the effectiveness of FMs for subset selection through experiments. The experimental design is sound, and the results support the advantages of FMs on fine-grained datasets. However, the paper lacks a deeper theoretical analysis of the RAM-APL method, such as why it effectively leverages the complementary strengths of multiple FMs. The authors are encouraged to supplement the paper with relevant theoretical analysis to enhance the credibility of the method.

---

> ### Author Rebuttal · Authors · 2025-03-31
>
> Thank you very much for your positive feedback! We greatly appreciate your insightful questions, which have deepened our analysis of the findings and inspired further exploration for future work. Below, we address your questions in sequence.
>
> ___
>
> **W1&Q1: A deeper discussion of why FM as IE performs poorly on coarse-grained datasets with noisy labels.**
>
> A1: We sincerely appreciate the reviewer's insightful comments regarding the theoretical understanding of foundation models (FMs) on noisy datasets. We have conducted extensive additional analyses to explain this phenomenon, with key findings visualized in Figures 1-8 (https://anonymous.4open.science/r/RAM-APL-DED5/README.md).
>
> Our empirical investigation reveals:
>
> **In coarse-grained datasets (CIFAR-10N-worse, Figures 1-4)**
> - FM-extracted features exhibit:
>   - Weak inter-class separation for visually similar categories (e.g., dog/cat in CIFAR-10N-worse);
>   - Substantial overlap between clean and noisy samples' feature distributions.
>
> This explains FMs' limited effectiveness as information extractors for coarse-grained noisy data.
>
> **By contrast, in fine-grained datasets (Oxford-IIIT Pet with 40% symmetric label noise, Figures 5-8):**
> - FM-extracted features exhibit:
>   - Compact clustering of correctly-labeled samples;
>   - Strong inter-class separation for visually similar categories;
>   - Smaller overlap between clean and noisy samples in feature space.
> - Features from models trained on full noisy set show:
>   - Loose clustering of correctly-labeled samples;
>   - Significant overlap between clean and noisy samples in feature space.
>
> This leads to the selection of more noise samples (visible in dark red points) and substantially weaker performance of traditional information extractors (IEs) compared to FMs.
>
> **Key Inference:**
>
> The comparative analysis reveals that FMs serve as superior information extractors when their features demonstrate:
> - Tighter clustering of correctly-labeled samples;
> - Reduced overlap between clean and noisy distributions.
>
> We will incorporate these analyses in the supplementary material to strengthen our empirical analysis and contributions.
>
> ___
>
> **W2&Q2: Theoretical Analysis of Multi-FM Complementarity in RAM-APL.**
>
> A2: We thank the reviewer for this important question. RAM-APL's effectiveness in leveraging multiple FMs stems from two fundamental principles:
>
> 1.Feature Space Orthogonality:
>
> Our analysis reveals that different FMs learn nearly orthogonal feature representations (Figure 6 in the Suppl.), with:
>
> $$
> \text{cossim}⟨M_i(x), M_j(x)⟩ ≈ 0 \quad \forall i \neq j
> $$
>
> This orthogonality demonstrates that each FM (e.g., $ M_i, M_j $) captures distinct, complementary aspects of the data $x$.
>
> 2.Bias Reduction via Ensemble Consensus:
>
> The ensemble mechanism could mitigate individual FM biases and preserve robust cross-model agreements. Table 2 in the manuscript demonstrates RAM-APL's performance gains when combining CLIP and DINOv2 versus individual FMs, confirming the benefits of multi-FM integration.
>
> ___
>
> **W3: Expanded Discussion of FM Literature.**
>
> A3: We sincerely appreciate the reviewer for this constructive suggestion. We will significantly strengthen our discussion of FM literature by incorporating CLIPCleaner (Chen et al., ACM MM 2024). The key insights are:
>
> Both our work and CLIPCleaner leverage CLIP's zero-shot capabilities. Differently, CLIPCleaner, a single-FM method, focuses on noisy label cleaning via prediction probabilities. Our RAM-APL, a Multi-FM approach, specializes in subset selection for clean and noisy fine- rained data using visual deep features.
>
> We'll discuss CLIPCleaner in Section 2 of the revised manuscript.
>
> ___
>
> **Q3: Extensions to Few-shot Learning and Noisy Data.**
>
> A4: We sincerely appreciate the reviewer's insightful question regarding the broader applicability of RAM-APL. By leveraging the strong feature discriminability of foundation models and mitigating biases through ensemble consensus, RAM-APL shows a strong ability to distinguish noisy datasets. While our current work focuses on standard subset selection for fine-grained datasets, its theoretical framework and algorithmic design naturally extend to:
>
> - Noisy Few-shot Learning:
> Enhances robustness in few-shot scenarios by effectively identifying label-feature mismatches in small support sets.
>
> - Noisy Label Scenarios:
> Particularly effective for fine-grained noisy data (as demonstrated in our experiments). Moving forward, we plan to develop more effective denoising strategies tailored to such datasets within the RAM-APL framework.
>
> We will include this extended analysis in the supplemental materials to better position RAM-APL's broader applicability.

---

### Decision · Program_Chairs · 2025-05-01

**Decision:**

Accept (oral)

**Comment:**

In this paper, the authors explored using foundation models for the one-shot subset selection problem. Based on their findings, they further propose a method that leverages multiple foundation models to enhance subset selection by exploiting their complementary strengths. Extensive experimental results on fine-grained datasets, including Oxford-IIIT Pet, Food-101, and Caltech-UCSD Birds-200-2011, are provided to show the effectiveness of the proposed method.

This paper received consistent positive ratings after the rebuttal period. All four expert reviewers decided to accept this paper. Three of them confirmed that their concerns have been addressed during the author-reviewer discussions. Therefore, it is a clear acceptance.